# Horizontal transfers between fungal *Fusarium* species contributed to successive outbreaks of coffee wilt disease

Lily D. Peck[1,2,3]*, Theo Llewellyn[1,4], Bastien Bennetot[5], Samuel O'Donnell[5], Reuben W. Nowell[6,7], Matthew J. Ryan[3], Julie Flood[3], Ricardo C. Rodríguez de la Vega[5], Jeanne Ropars[5], Tatiana Giraud[5], Pietro D. Spanu[8], Timothy G. Barraclough[2,6]

1 Science and Solutions for a Changing Planet, Grantham Institute, Imperial College London, London, United Kingdom, 2 Department of Life Sciences, Silwood Park Campus, Imperial College London, Berkshire, United Kingdom, 3 CABI, Egham, Surrey, United Kingdom, 4 Comparative Fungal Biology, Royal Botanic Gardens, Kew, Richmond, United Kingdom, 5 Ecologie Systematique et Evolution, CNRS, AgroParisTech, Université Paris-Saclay, Gif-sur-Yvette, France, 6 Department of Biology, University of Oxford, Oxford, United Kingdom, 7 Biological & Environmental Sciences, University of Stirling, Scotland, United Kingdom, 8 Department of Life Sciences, Sir Alexander Fleming Building, Imperial College London, United Kingdom

* l.peck18@imperial.ac.uk, ldpeck@ucla.edu

**Data Availability Statement:** Improved assemblies and raw sequencing data have been deposited in the relevant International Nucleotide Sequence

## Abstract

Outbreaks of fungal diseases have devastated plants and animals throughout history. Over the past century, the repeated emergence of coffee wilt disease caused by the fungal pathogen *Fusarium xylarioides* severely impacted coffee production across sub-Saharan Africa. To improve the disease management of such pathogens, it is crucial to understand their genetic structure and evolutionary potential. We compared the genomes of 13 historic strains spanning 6 decades and multiple disease outbreaks to investigate population structure and host specialisation. We found that *F. xylarioides* comprised at least 4 distinct lineages: 1 host-specific to *Coffea arabica*, 1 to *C. canephora* var. robusta, and 2 historic lineages isolated from various *Coffea* species. The presence/absence of large genomic regions across populations, the higher genetic similarities of these regions between species than expected based on genome-wide divergence and their locations in different loci in genomes across populations showed that horizontal transfers of effector genes from members of the *F. oxysporum* species complex contributed to host specificity. Multiple transfers into *F. xylarioides* populations matched different parts of the *F. oxysporum* mobile pathogenicity chromosome and were enriched in effector genes and transposons. Effector genes in this region and other carbohydrate-active enzymes important in the breakdown of plant cell walls were shown by transcriptomics to be highly expressed during infection of *C. arabica* by the fungal arabica strains. Widespread sharing of specific transposons between *F. xylarioides* and *F. oxysporum*, and the correspondence of a putative horizontally transferred regions to a *Starship* (large mobile element involved in horizontal gene transfers in fungi), reinforce the inference of horizontal transfers and suggest that mobile elements were involved. Our results support the hypothesis that horizontal gene transfers contributed to the repeated emergence of coffee wilt disease.

Database Collaboration (INSDC) database with the BioProject ID PRJNA1043203 and the SRA run accession SRP491020 (see S3 File). This study also analysed publicly available data from accession numbers described in S2 Table. The exact versions of genome assemblies and annotations analysed in this manuscript are published in a publicly available Zenodo digital repository (https://doi.org/10.5281/zenodo.13836286), along with source data, custom analysis and plotting scripts, and bioinformatic workflows used in this study.

**Funding:** This research was supported by the Natural Environment Research Council [Grant number NE/L002515/1 (LDP) and NE/S010866/1 (TGB).] The funders had no role in study design, data collection and analysis, decision to publish, or preparation of the manuscript.

**Competing interests:** The authors have declared that no competing interests exist.

**Abbreviations:** FDR, false discovery rate; gVCF, genomic variant call format; HGT, horizontal gene transfer; HTR, horizontally transferred region; INSDC, International Nucleotide Sequence Database Collaboration; ML, maximum likelihood; SIX, secreted in xylem; SNP, single-nucleotide polymorphism; TE, transposable element.

## Introduction

Fungal diseases have devastated plants and animals throughout history, with the first records dating from biblical plagues [1]. Many cases of famines and harvest failures resulted from outbreaks of fungal diseases with wide-ranging and important human consequences, with between 10% and 23% yields lost to fungal disease, and modern farming practices risk inadvertently helping the emergence of new pathogens [2]. Coffee is a major commodity that is worth over $22 billion to the global economy [3], provides high export earnings (e.g., one third exports in Ethiopia [4]) and supports millions of smallholder farmers [5]. However, as with other crops, there is a constant threat from pests and disease, especially fungal pathogens. Coffee wilt and leaf rust have caused major losses to crop yields and livelihoods in Africa and Central America, respectively [6,7]. Pathogens can evolve rapidly [2], so understanding how new variants emerge is vital to developing sustainable disease control measures.

Coffee wilt disease, caused by the fungal pathogen *Fusarium xylarioides*, infects coffee plants via roots or wounds to colonise the xylem [8]. Wilting results from the pathogen blocking water uptake, and cell walls are degraded for nutrients [9]. Currently, 2 host-specialised strain types are known to infect the main African coffee crops: arabica (*Coffea arabica*), grown at high altitudes, and robusta (a variety of *C. canephora*), grown at low-mid altitudes in tropical Africa [8]. Coffee wilt disease has a well-documented history of emergence and outbreaks since its discovery in 1927 [6]. By the 1950s, coffee crops in west and central Africa were decimated, resulting in the replacement of commercial cultivars with resistant robusta coffee [6]. Yet by the 1970s, coffee wilt disease re-emerged on robusta coffee in central Africa [10], becoming a major problem in east and central Africa by the mid-1990s. Production did not recover in some countries (e.g., Democratic Republic of Congo, [11]). In parallel, coffee wilt disease was first reported on arabica coffee in Ethiopia in the 1950s, becoming widespread in the 1970s, but with low severity of disease [12]. However, one recent study suggests that disease severity in Ethiopia has increased [13]. Key questions for understanding the emergence of coffee wilt disease include: (i) what are the genetic relationships among the strain types on different hosts and at different times; and (ii) what genetic changes underlie their ability to infect different hosts?

There is growing evidence that horizontally transferred regions (HTRs) between distantly related fungi are important for the emergence of new pathogen variants [14]. For example, a 14 kb fragment encoding the ToxA virulence protein and abundant transposons was transferred between 3 wheat fungal pathogens of the order *Pleosporales* [15]. This fragment was later identified as being part of a *Starship* [16], a newly described group of large mobile elements found across every major class of filamentous ascomycetes that can conflict or cooperate with their fungal host genome [17]. *Starships* mediate the movement of large sets of genes within genomes, species, or even between species through horizontal transfer. They were recently linked to genetic and phenotypic variability in a human pathogen [18]. In addition to genomic evidence, there has been experimental evidence of *Starships* as autonomous transposons [19].

Interspecies transfers can also occur via mobile accessory chromosomes. For example, *F. oxysporum*, which causes wilt disease in over 100 plant species with host-specific *formae speciales* [20], contains supernumerary chromosomes which transfer between *formae speciales* and determine host specificity [21–23]. These chromosomes contain 4 named "supercontigs," subregions enriched in secreted in xylem effector genes (*SIX*) and *mimp* transposons. *SIX* genes encode small proteins secreted by *F. oxysporum* into the xylem of a host plant during infection and which promote virulence [24], while *mimp* transposons are a type of nonautonomous miniature-inverted *impala* repeat which are always present in the promoter of *F.*

*oxysporum SIX* genes and consequently associated with virulence [25]. Transposons are hypothesised to promote hyper-variability of sets of effector genes, either directly by transposition or by facilitating recombination to generate new gene assortments [26]. Thus, while core fungal genomes may generally evolve separately within local species gene pools, transfers between more distant relatives can introduce diversity and enable outbreaks of new types of pathogen [27].

A previous study sequenced 6 *F. xylarioides* genomes comprising arabica, robusta, and strains isolated in 1950s which we describe as "coffea" for their ability to infect multiple *Coffea* species [28]. Genome-wide variation supported an early split between arabica and other strain types, consistent with pre-agricultural divergence on wild coffee species, and potentially a separate gene pool on *C. arabica*. In contrast, the robusta strain type from the 1970s onwards appeared to derive from the pre-1970s coffea strain type outbreak. Different strain types featured different putative effector genes, whereby some effectors showed close matches with *F. oxysporum*, which were lacking in closer *F. xylarioides* relatives. Strikingly, a 20-kb scaffold showed high sequence similarity to an *F. oxysporum* pathogenicity chromosome and contained the same genes and *mimp* transposons, while these were lacking in closer *F. xylarioides* relatives. This led to the hypothesis that host specialisation and changes in subsequent outbreaks involved divergent strain types receiving HTRs from several *F. oxysporum* formae speciales (ff. spp.) and thus a greatly expanded gene pool of wilt-associated effector genes.

Here, we test this hypothesis using comparative genomics of 11 *Fusarium* genomes, including *F. oxysporum* and *F. solani* chosen to represent potential HTR donors in crops that grow with or near coffee in Africa (S1 Table). Aligning *F. xylarioides* genomes with a new long-read reference genome for the arabica strain type, we found that the different outbreaks on diverse coffee types arose in separate populations. The populations differ in the presence and absence of large genomic regions that, when present, show high sequence similarity with multiple different *F. oxysporum* ff. spp. compared to genome-wide similarity, but are found in non-homologous locations in genomes, contrasting with an otherwise high synteny level; this is consistent with horizontal acquisition from distantly related lineages. Transcriptomics revealed that putative effector genes were highly expressed during coffee plant infection, with several putatively originating from *F. oxysporum* and absent in closely related species to *F. xylarioides*. Finally, all *F. xylarioides* populations share highly similar *mimps* and other transposable elements associated with *F. oxysporum*'s mobile pathogenic chromosome, and, notably, a *Starship* element, which could have mediated horizontal gene transfer (HGT) between these 2 species. Together, these results support the hypothesis that HGT contributed to the emergence of host-specialised populations that caused successive outbreaks of coffee wilt disease.

## Results

### A high-quality genome assembly for the *Fusarium xylarioides* arabica host specialist

Assembly of Nanopore long-read data for *F. xylarioides* IMI 389563 ("arabica563") generated its first high-quality reference genome with 21 contigs, including 12 contigs over 1 Mb, and 4 contigs over 100 kb. Telomeric repeats were identified on both ends of 3 chromosomes, consistent with these representing complete assembled chromosomes, and at just one end of 7 contigs, probably representing incomplete chromosomes (S5B Fig). The assembly length was 57 Mb (with 55 Mb in the 12 longest contigs) with an N50 greater than 5 Mb (Table 1). The Rag-Tag assemblies ordered and oriented to the reference genome for the remaining *F. xylarioides* strains allowed the identification of various large arabica-specific regions over 0.7 Mb in

**Table 1. Statistics for sequenced *Fusarium* genomes.** Arabica563 is the reference assembly. Only contigs >1 kb in length were included. *Fusarium oxysporum* is abbreviated to *Fo*.

| Strain name | Strain ID | Genome size | No. contigs | N50 contig length | Genome completeness | Genome GC | Coding genes |
|---|---|---|---|---|---|---|---|
| | (IMI) | Mb (>1 kb) | n (> 1kb) | n (> 1kb) | BUSCO (%) | % | n |
| Arabica563 | 389563 | 57.3 | 21 | 5,156 | 100.0 | 42.2 | 17,424 |
| Arabica038 | 507038 | 51.1 | 4631 | 24.5 | 97.9 | 43.4 | 15,705 |
| Robusta268 | 392268 | 50.2 | 4420 | 27.1 | 97.6 | 44.1 | 15,575 |
| Coffea035 | 507035 | 51.1 | 3880 | 30.4 | 97.9 | 43.9 | 15,588 |
| Coffea113 | 507113 | 48.7 | 3795 | 30.1 | 98.3 | 44.3 | 15,115 |
| Coffea676 | 392676 | 50.0 | 4362 | 27.1 | 97.9 | 44.4 | 15,649 |
| *F. solani* 280 | 392280 | 48.4 | 1867 | 45.2 | 96.9 | 50.9 | 14,007 |
| *Fo* 509 | 244509 | 49.7 | 1923 | 67.9 | 98.3 | 47.4 | 14,475 |
| *Fo cubense* 109 | 141109 | 48.4 | 1992 | 60.1 | 99.0 | 47.6 | 14,993 |
| *Fo pisi* 221 | 500221 | 52.9 | 4299 | 28.1 | 95.5 | 47.5 | 16,614 |
| *Fo raphani* 541 | 337541 | 53.8 | 4285 | 38.1 | 98.3 | 47.7 | 16,421 |
| *Fo vasinfectum* 248 | 292248 | 49.4 | 1904 | 72.5 | 98.6 | 47.4 | 14,846 |

contigs 6, 7, 8, 12, and 13, compared to robusta-specific genomes. The regions in contigs 7, 12, and 13 were absent from all robusta- and coffea-specific genomes.

## *Fusarium xylarioides* is a monophyletic species complex with genetically differentiated populations

Long-read sequencing of a new *F. xylarioides* reference genome revealed the same core chromosomes as *F. verticilliodes* (Fig 1A), another species of the *Fusarium fujikuroi* species complex (Fig 1C). Short-read genome sequencing of *F. xylarioides* strains isolated across the spatial and temporal range of coffee wilt disease (Fig 1B) supported the delimitation of *F. xylarioides* as a species complex (Figs 1C and S1). Analysing 11 new Illumina genome assemblies of 5 *F. xylarioides* strains and 6 other *Fusarium* genomes (S1 Table and Fig 1), together with 30 published *Fusarium* genomes (S2 Table), found 26,000 orthologous groups of protein-coding genes, of which 3,544 had members in all genomes with 1,685 single-copy genes. Two species trees, a "supermatrix" tree reconstructed from single-copy gene trees (Figs 1C and S1), as well as a "supergene" tree reconstructed from the 3,544 orthologous gene trees (S2 Fig), supported phylogenetic relationships from earlier studies with lower sampling [28,29]. *Fusarium xylarioides* forms a monophyletic clade within the *F. fujikuroi* species complex, a result that receives 100% bootstrap branch support in concatenated analysis of single-copy genes (a "supermatrix" tree, 1,000 replicates, Fig 1C) and support from 58% of genes in the "supergene" tree (branch support values, S2 Fig). Average nucleotide identities in coding sequences within *F. xylarioides* (98%) are high compared to lower identities with other species (*F. xylarioides* to *F. phyllophilum* 96%, to the *F. fujikuroi* complex 91% to 92%, and to *F. oxysporum* strains 90%, S3 Fig). Multilocus species delimitation based on patterns of congruence/incongruence among gene trees (using the tr2 algorithm) identified significant units that matched named species except for excluding one strain from *F. oxysporum* (*F. oxysporum rapae* Tf1208 isolate, Fig 1C). Within *F. xylarioides*, 4 sub-clades were present: arabica, robusta, and 2 separate clades of coffea genomes (differentiating those isolated from the pre-1970s outbreak, strains 659 and 676, from the others) (Figs 1C and S1). Across all orthogroups present in these sub-clades, 80% genes are shared by all clades (S4 Fig), yet relationships were not highly congruent across gene trees, with branch support values <60% in the "supergene" tree (Fig 1C). This lack of congruence may reflect incomplete lineage sorting due to short internode intervals.

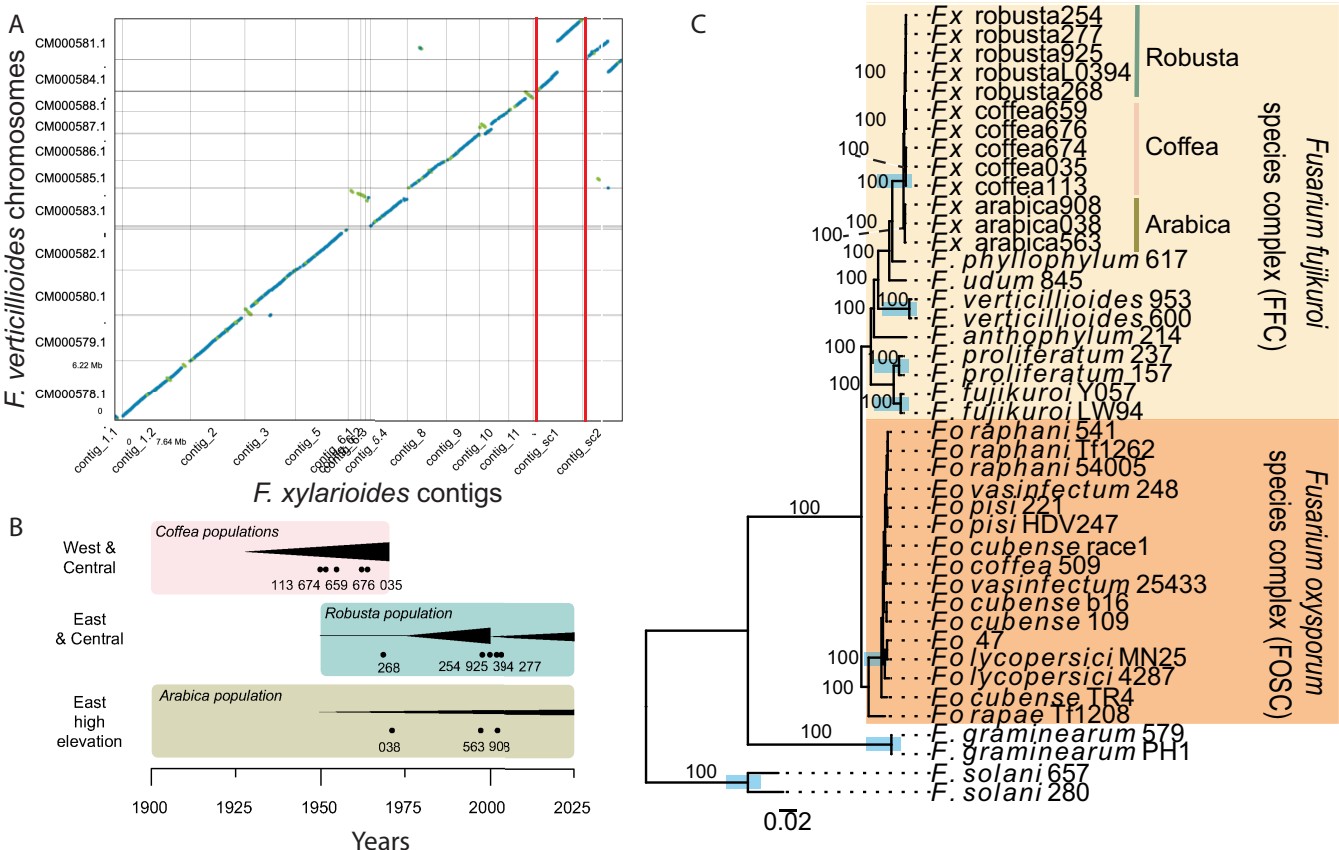

**Fig 1. The *Fusarium xylarioides* species complex.** (A) Dotplot visualisation of whole-genome alignment between the *Fusarium xylarioides* arabica563 reference genome and *F. verticillioides* chromosomal assembly (GCA_000149555.1). Orange lines indicate chromosomal translocations; blue dots indicate forward and green dots reverse alignments; gaps indicate no alignment. (B) Schematic summary of geography and history of outbreaks and sampling times of isolates. Boxes = main coffee crops. Black triangles = qualitative incidence of coffee wilt disease. Dots = sample times for our isolates, labelled with the last 3 numbers of each *F. xylarioides* strain number (full strain details are in S1 and S2 Tables). Schematic not to scale: incidence estimated from [6,11,13,30]. (C) Gene sharing across the *Fusarium xylarioides* host-specific populations. Species tree based on 1,685 concatenated single-copy orthogroups. Annotated branches indicate bootstrap branch support values and blue boxes indicate resolved monophyletic species clades identified by multilocus species delimitation. Shaded boxes refer to the *F. fujikuroi* (yellow) and *F. oxysporum* species complex (orange). The scale bar indicates branch length (0.02 substitutions per site). The *F. xylarioides* populations arabica, robusta, coffea1, and coffea2 are labelled. Fx, *F. xylarioides*; Fo, *F. oxysporum*. Full strain details in S1 and S2 Tables. The data underlying this figure can be found in https://doi.org/10.5281/zenodo.13836286.

Analyses of genetic variation within *F. xylarioides* revealed population subdivision consistent with the hypothesis that multiple outbreaks arose from separate gene pools. A Neighbour-Net analysis based on 239,944 single-nucleotide polymorphisms (SNPs) from coding and noncoding regions suggests 4 differentiated populations (Fig 2, absolute divergence $d_{XY}$ >0.001 for all pairwise comparisons, S3 Table): an arabica group, a robusta group, and the same 2 coffea clades as recovered on the species tree (Fig 1C). A small amount of reticulation occurs among strains in the robusta group, indicating current or ancient gene flow. There is no reticulation between the arabica group and either the robusta or coffea groups, suggesting a lack of gene flow and recombination between the arabica clade and the other clades.

## The populations differ in the presence of multiple large genome regions

We identified numerous large genomic regions (search size 10 kb) that varied in their presence/absence between the *F. xylarioides* arabica genomes and the other *F. xylarioides* populations (presence defined as >50% similarity after whole-genome alignment using minimap2,

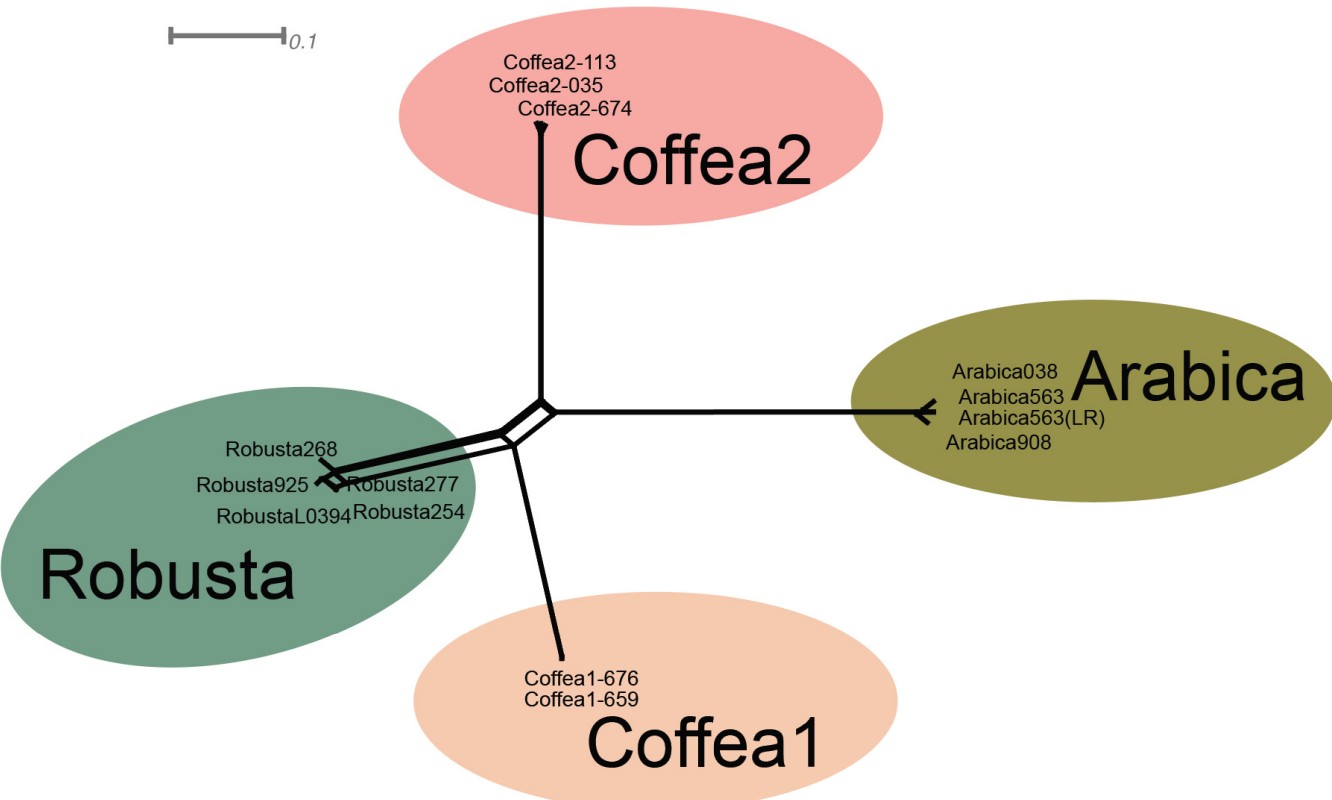

**Fig 2. Genetic clusters in *Fusarium xylarioides*.** A Neighbour-Net (SplitsTree) analysis based on a single-nucleotide polymorphism distance matrix. The scale bar represents 0.1 substitutions per polymorphic site. The points are shaded according to their host population. The data underlying this figure can be found in https://doi.org/10.5281/zenodo.13836286.

see Methods). In total, 2.35 Mb were differentially present among the *F. xylarioides* populations, with nearly half (1.7 Mb) specific to the arabica population, comprising 70 mostly-contiguous regions (namely a gap of 20 kb between regions is still considered contiguous) of 10 kb or more on all chromosomes (Fig 3A). The longest contiguous arabica-specific region was a 360 kb region of chromosome 1 (contig 13 in S5 Fig). Together, these regions contained 1,071 predicted genes and 2,623 annotated repeats. We therefore hypothesised that the gain or loss of genome regions could underpin specialisation of the different *F. xylarioides* host populations.

### Several population-specific genome regions show close matches to pathogenicity chromosomes of multiple *Fusarium oxysporum* lineages

We compared population-specific genome regions to all genomes in the NCBI nonredundant ("nr") database and found that 9 contiguous large genomic regions (>20 kb) were also present in *F. oxysporum*. Of those regions, 5 were absent from other *F. fujikuroi* complex genomes and with unexpectedly strong BLAST hits to various *F. oxysporum* lineages ($Evalue < 10^{-20}$, Figs 3B, 3C, S6A–S6C and 4.) An additional region was specific to the robusta and coffea genomes but with a strong BLAST hit to *F. oxysporum* f. sp. *lycopersici* ($E$-value $10^{-169}$, S6D and S7 Figs). We hypothesised that these could be putative HTRs shared by *F. xylarioides* and *F. oxysporum*, which we named HTR 1 to 6. They range from 20 kb (HTR 4) to 200 kb (HTR 1). Putative HTRs 1, 3, and 4 were found only in the arabica population; HTR 2 was found in arabica,

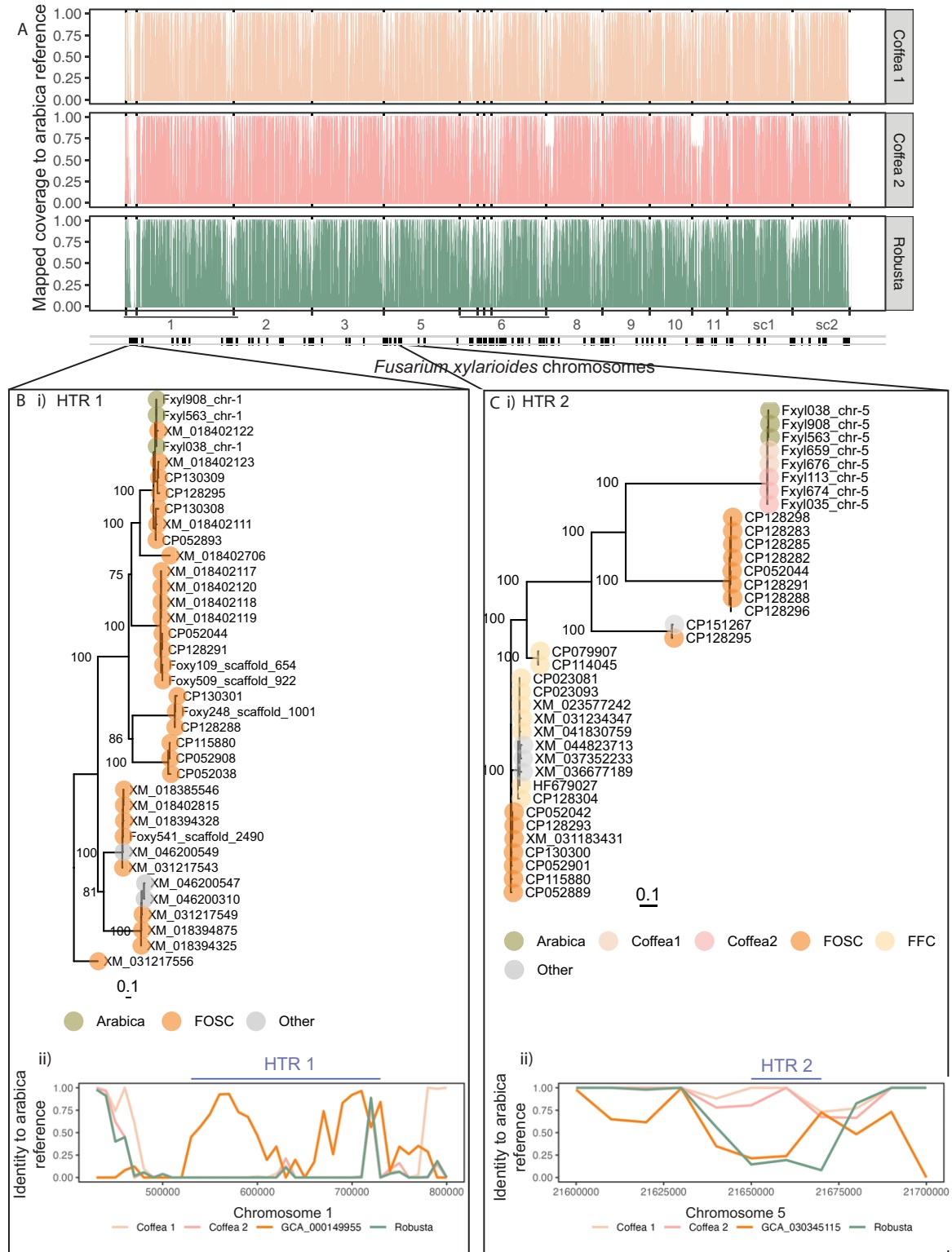

**Fig 3. Host-specific genome regions between members of the *Fusarium xylarioides* species complex comprise putative horizontally transferred regions.** (A) The genomes of the *F. xylarioides* coffea1, coffea2, and robusta populations compared to the arabica563 reference show high coverage. Y-axis shows coverage (fraction of bases covered) from minimap2 (-cx asm10 --secondary = no --cs) mapping between the arabica reference and each coffea1 (coloured peach, top bar), coffea2 (coloured pink, middle bar), and robusta (coloured

turquoise, bottom bar) genomes (data plotted as average of 100 × 100 bp = 10 kb windows). See S1 and S2 Tables for strain details, vertical black bars in each plot denote the original contigs, see S12 Table for contig names. However, some regions are differentially present or absent between populations, annotated with black boxes in the bar below the x-axes (71 in total comprising 2.35 Mb). These indicate the *F. xylarioides* host-specific genome regions, namely: those present only in arabica; in arabica and robusta; in arabica and coffea1; in arabica and coffea2. sc1, supercontig 1; sc2, supercontig 2. (B i) Unrooted ML tree of HTR 1 and matching sequences from different *Fusarium* genomes. A BLAST search against the "nr" database with the *F. xylarioides* HTR 1 sequence as query recovered its sequence in 3 *Fusarium* species: *F. xylarioides* arabica (green tip labels), *F. oxysporum* (orange tip labels), and *F. redolens* (grey tip labels). The ML tree shows high bootstrap support (100% from 1,000 bootstraps) for clustering of the HTR 1 sequences of *F. xylarioides* arabica with that of *F. oxysporum*. The HTR 1 sequence is absent from other *F. xylarioides* populations and *F. fujikuroi* complex species. Bootstrap support for all internal nodes with a support value >70% are shown. Tip labels correspond to NCBI GenBank accession numbers. *FOSC*, *F. oxysporum* species complex. The scale bar indicates branch length (number of substitutions per site). (B ii) Sliding window plot of genome similarity between *F. xylarioides* host-specific populations and *F. oxysporum* as in Fig 3A but zooming in on the HTR 1 region with the addition of *F. oxysporum* f. sp. *lycopersici* GCA_000149955 (orange line) that shares a highly supported clade with *F. xylarioides* (XM 018402122). The location of HTR 1 is indicated with a blue line and is absent from coffea1, coffea2, and robusta populations (each line is at 0 on the y-axis), whereas it is present and similar in *F. oxysporum* f. sp. *lycopersici* GCA_000149955 (2 peaks with >90% similarity). (C i) Unrooted ML tree of HTR 2 from different *Fusarium* genomes. Same as above, but HTR 2 was recovered in *F. xylarioides* arabica, coffea1, and coffea2 populations (green, peach, and pink tip labels, respectively), *F. oxysporum* (orange tip labels), and *F. fujikuroi* species complex genomes (yellow tip labels, *F. verticillioides*, *F. mangiferae*, *F. fujikuroi*, *F. proliferatum*) and distantly related *Fusarium* species (grey tip labels). (C ii) Sliding window plot of genome similarity for HTR 2. Same as above, but with the addition of *F. oxysporum* Fo5176 GCA_030345115 (orange line) because this genome shares a highly supported clade with *F. xylarioides* (CP128298, CP128283, CP128285, CP128282, CP128291, CP128288, CP128296). The data underlying this figure can be found in https://doi.org/10.5281/zenodo.13836286.

coffea1, and coffea2 populations; HTR 5 was found in all *F. xylarioides* populations; and HTR 6 in robusta, coffea1, and coffea2 populations. HTR 2 (in arabica and coffeas) and HTR 6 (in robusta and coffeas) showed strong matches to different parts of "supercontig 51" from the mobile pathogenic chromosome in *F. oxysporum* f. sp. *lycopersici*. For HTRs 1 to 4, the mean sequence identity to closest matches in other *F. xylarioides* strains (denoted $I\ Fx$) was 5.8%, 5.8%, 4.7%, and 0%, respectively, whereas the mean identity to closest *F. oxysporum* matches ($I\_Fo$) was 55.2%, 39.3%, 84.4%, and 88.5%. To test if these values were statistical outliers compared to the genome-wide average, we aligned similarly sized windows to both *F. xylarioides* and *F. oxysporum* genomes and compared the observed values of the difference ($I\_Diff = I\_Fx - I\_Fo$) to this background distribution. The proportion of background windows with an $I\_Diff <$ the observed value was 0.018%, 6.7%, 0.25%, and 0.54% for HTRs 1 to 4, respectively, indicating that most HTR regions were extreme outliers compared to the background distribution of sequence identity across *F. xylarioides* and *F. oxysporum* genomes (S13 Table).

Phylogenetic analyses on alignments of each whole putative HTR provided further support for the horizontal transfer hypothesis. In every case, *F. xylarioides* grouped with *F. oxysporum* genomes with strong bootstrap support (all >99%). In 5 cases (HTRs 1, 3–6), the putative HTRs were not recovered from any species from the *F. fujikori* complex (Figs 3B and S6), which should be closer phylogenetic relatives of *F. xylarioides* (Fig 1C). We report a close relationship between HTR 4 and *F. oxysporum* 509 (S6D Fig), isolated from a coffee tree and sequenced in this study, as well as *F. oxysporum* f. sp. *pisi*, also sequenced here, which clustered with *F. xylarioides* arabica nodes for HTR 5 (S6C Fig). To test if pairwise divergence between the putative HTRs and the nearest *F. oxysporum* ($D\_FxFo$) copy were statistical outliers compared to background levels of divergence, we compared putative HTR gene trees with the nearest *F. oxysporum* copy ($D\_FxFo$), and then compared the observed values of the difference with the metric of divergence in single-copy gene trees. The proportion of HTRs with a $D\_FxFo \leq$ the observed value was: HTR 1, 9.4%; HTR 3, 1.1%; HTR 4, 4.3%; HTR 5, 0.13%; HTR 6, 0.84%, S14 Table. In HTR 2, the same genome region was recovered from one or more *F. fujikori* complex species, but they clustered with different *F. oxysporum* copies in the tree (Figs 3C, S5A and S5B). Pairwise divergence between the *F. xylarioides* and the closest *F. oxysporum* match ($D\_FxFo$) was unexpectedly lower than the divergence between *F. xylarioides* and the closest species from the *F. fujikori* complex species match ($D\_FxFFC$), relative to the

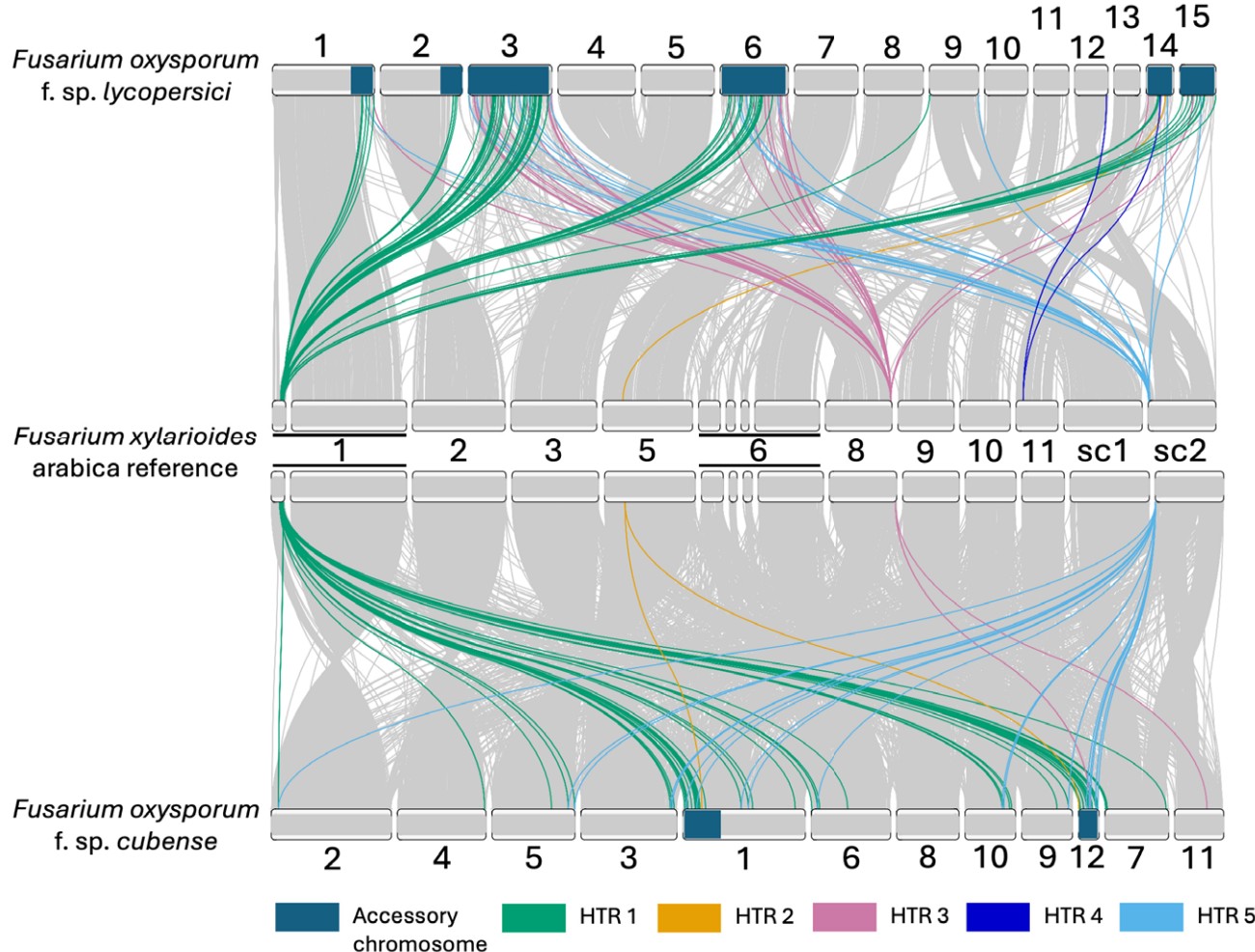

**Fig 4. Synteny between the *Fusarium xylarioides* arabica reference genome and 2 *Fusarium oxysporum* genomes for the putative HTRs.** Long-read assemblies for each genome were used (GCA_000149955.2 for *F. oxysporum* f. sp. *lycopersici* and *F. oxysporum* f. sp. *cubense* [31]). Nucmer whole genome alignments were visualised in RIdeogram. The links indicate matching blocks larger than 200 bp, coloured in grey. Accessory chromosomes, as described by [21,31], are coloured in turquoise. The HTRs are shown on the chromosomes by colours as indicated on the figure key.

expected levels (percentile of $D\_FxFo < D\_FxFFC$) across single copy ortholog trees: 0.0%. In all cases, *F. oxysporum* sequences were highly divergent with long branch lengths in maximum likelihood (ML) trees in Figs 3 and S6, whereas *F. xylarioides* displayed low variation and short/identical branch lengths in the same ML trees.

The putative HTRs all include BLAST hits to extremities of mobile and pathogenicity chromosomes known to be associated with host specialisation in *F. oxysporum* formae speciales (Fig 4), but never to a single chromosome. For example, the largest HTR (HTR 1) is on chromosome 1 (contig 1.1 in Fig 1A) in *F. xylarioides* arabica strains, in a region which is absent from the *F. verticillioides* chromosome assembly (Figs 1A and S5A). One half of contig 1.1 is found in all strains (e.g., Figs 1A and S5B), albeit with very low sequence identity outside *F. xylarioides*. The other half, which mostly comprises HTR 1, is unique to the *F. xylarioides* arabica population and *F. oxysporum*, with high identity (>92%) in relatively short pieces (*F. oxysporum* f. sp. *vasinfectum* CP130309, 22.6 kb; *F. oxysporum* Fo5176 CP128295, 19.2 kb; *F. oxysporum* f. sp. *lycopersici* XM_018402123, 7 kb) compared to the total 150 kb putative HTR

region (Fig 3C). Compared to the best-assembled *F. oxysporum* f. sp. *lycopersici* and *F. oxysporum* f. sp. *cubense* genomes, a synteny alignment reveals that HTR 1 matches in part to all of their mobile chromosomes, namely chromosomes 1, 3, 6, 14, and 15 in *F. oxysporum* f. sp. *lycopersici* and chromosomes 1 and 12 (along with other non-accessory regions) in *F. oxysporum* f. sp. *cubense* (Fig 4). Within HTR 1, there are 39 genes with a high sequence identity between the *F. oxysporum* f. sp. *lycopersici* and *F. xylarioides* arabica strains (BLAST, >94% nucleotide identity compared with 88% identity in flanking regions and whole genome identity of 90%, S3 and S8 Figs). Two of these genes belong to orthologous groups only found in *F. oxysporum* and the *F. xylarioides* arabica strains, with highly similar *F. xylarioides* arabica copies nested within diverse *F. oxysporum copies* from various *formae speciales* (S8B and S8C Fig).

As a second example, HTR 2 comprises a 30-kb region on chromosome 5 (Figs 3C and 4). It contains 3 genes shared by all *F. xylarioides* arabica and coffea populations (arabica563 gene copies are H9Q71 0016558, H9Q71 0017194, H9Q71 0017193) and which match to *F. oxysporum* effector proteins: Six7 with 91% identity, Six10 with 92% identity, and Six12 with 94% identity (S7 Fig and S4 Table). Other *SIX* effector genes (*six1*, *six2*, *six6*, *six7*, and *six11*) were identified in members of the *F. fujikuroi* species complex, with vertical (for *six2*) and horizontal inheritance from the *F. oxysporum* species complex suggested [32]. In addition, we previously identified *six7* and *six10* in arabica and coffea populations [28]. There were no significant BLAST hits in the intergenic regions of HTR 2 outside *F. xylarioides*. Each *SIX* gene either contains a *mimp* in its gene body (*six10* and *six12*) or in its promoter region (*six7*, *mimp* 943bp upstream) (S7 Fig). In *F. oxysporum* f. sp. *lycopersici*, these 3 genes are found in supercontig 51 on the mobile, pathogenic chromosome, a region enriched in effector genes and *mimps*, among other transposable elements [25]. Remarkably, HTR 6, in the robusta and coffea populations, is shared with a different part of *F. oxysporum* f. sp *lycopersici*'s supercontig 51 to that shared with HTR 2 (Figs 3C, S6D, and S7). In each genome, HTR 6 comprises a single scaffold (approximately 20 kb): All genes in HTR 6 are unique to the *F. xylarioides* robusta and coffea populations and various *F. oxysporum* ff. spp., as identified by BLAST (S6D Fig). This region does not contain *SIX* effectors, but other genes important in infection, including a glycosyltransferase, Cytochrome P450 monooxygenases, a squalene-hopene-cyclase, a methyl-transferase-UbiE protein, and a Tri7 homolog.

### Putative horizontally transferred regions contain genes that are highly expressed during infection, including effector genes from *Fusarium oxysporum*

Among the expressed genes and relative to their relevant controls, an average of 2,164 were differentially expressed across the arabica563 samples, 1,713 across the arabica908 samples, and 233 across the coffee plant control samples. In the fungal samples, over half of the differentially expressed genes were up-regulated in vivo infection samples relative to controls, while in the coffee plant samples 80% of the differentially expressed genes (116 in the plants infected with arabica563; 231 with arabica908) were up-regulated in the infected samples. Genes which were differentially expressed in both *F. xylarioides* arabica strains belong to nearly 700 orthologous groups. The differentially expressed coffee genes are described in S10 Table.

To determine whether genes found on putative HTRs are associated with coffee wilt infection, we analysed up-regulated genes in planta during infection. Diverse carbohydrate-active enzyme (CAZyme) families capable of breaking down cellulose and pectin from plant cell walls, and effectors known in *F. oxysporum* such as the *SIX* genes, were previously identified in *F. xylarioides* arabica and robusta as putative effectors [28], but their expression during infection was unexplored. Analysis of gene expression of 2 strains of the *F. xylarioides* arabica clade

from infected arabica coffee plants, compared with the same strains grown *in axenic* liquid culture, revealed significant up-regulation in planta of genes encoding CAZyme families important in the breakdown of pectin, xylan, and other plant cell wall polymers (Fig 5A and 5B). In both strains, 7 pectin lyases (within the polysaccharide lyase PL family) were previously predicted as effectors: 3 PL1 genes; 3 PL3 genes including the pelA and pelD effectors characterised by [33]; and 1 PL9 gene (S5 Table). Most of the 23 pectin lyase genes from the subfamilies PL1, PL3, PL4, PL9, and PL26, showed strong up-regulation with 80% differentially expressed and 50% making up the top 20% of expressed genes (see S8 and S9 Tables for details).

Several putative effectors previously identified [28] were found to be significantly up-regulated in *F. xylarioides* from infected coffee plants, compared with fungi cultured in vitro (S6 and S7 Tables). Six putative effectors known from the mobile pathogenic chromosome of *F. oxysporum* f. sp. *lycopersici* were up-regulated in at least 1 arabica strain. These were: H9Q71 0011695 which appears to be a divergent type of *six7* effector (referred to as OG0014398 in [28]); 3 novel effector candidates identified by [25] which are all secreted proteins—unidentified *FOXG 14254*, oxidoreductase *orx1*, and a catalase-peroxidase found in infected xylem sap *FOXG 17460*; *sge1*, the *SIX* effector gene transcription factor; H9Q71 0017133, a LysM effector (referred to as OG0013477 in [28]); and 2 pectin lyase effectors, *pelA* and *pelD*. Only 1 putative effector, H9Q71 0014367, was expressed less in infected plants than control samples in both strains cultured in vitro. Seven remaining putative effectors were expressed but not differentially in both strains; 6 genes were expressed but not differentially in 1 strain (S6 and S7 Tables and S9 Fig). In addition to these putative effectors, reanalysis of the new reference genome with EffectorP 3.0, a predictor of apoplastic and cytoplasmic effectors in fungi [34] detected further pectinolytic enzymes, similar extracellular degradative enzymes, as well as the established fungal effector domains Ecp2 and LysM (see S5 Table for details), which were also up-regulated in infected plants.

There were 496 genes significantly up-regulated in planta in both arabica strains, of these, 19 were absent from *F. fujikuroi* complex species and mostly did not match any recognised domains (S4 Table). Of these 19 genes, remarkably, 3 were found in HTR 1 and were shared by *F. xylarioides* arabica and coffea populations only (H9Q71 0016558, H9Q71 0017194, H9Q71 0017193). Each matches to an *F. oxysporum* effector protein: Six7, Six10, and Six12, respectively (S4 Table). The *SIX* orthologues were also in the top 20% most highly expressed genes in planta across the arabica strains (Fig 5B). Of the remaining genes absent from *F. fujikuroi* complex species, several were population specific; some were shared by all *F. xylarioides* populations; and some were shared between *F. xylarioides* and other species complexes (S4 Table).

## Miniature *impala* elements and other transposons are shared between *Fusarium xylarioides* and *Fusarium oxysporum*

Certain transposable element (TE) families have been implicated in promoting effector gene variability and mobility in *F. oxysporum* and previously in *F. xylarioides* [25,28,32], and so we surveyed for the same transposon classes in the new *F. xylarioides* genomes. A recent study found one or a few *mimps* in 3 *F. fujikuroi*- and one *F. tricinctum* complex species, and linked their presence to horizontal transfer [32]. *Fusarium xylarioides* genomes contained miniature *impala* elements (*mimps*) that are present in *F. oxysporum* genomes but lacking in more closely related species, as well as other transposons previously identified in mobile *F. oxysporum* chromosomes. These elements are present in some putative HTRs and close to or inside several highly expressed genes. A total of 450 full-length *mimps* were detected across the *F. xylarioides* and *F. oxysporum* genomes (Fig 6A). *Fusarium oxysporum* formae speciales genomes

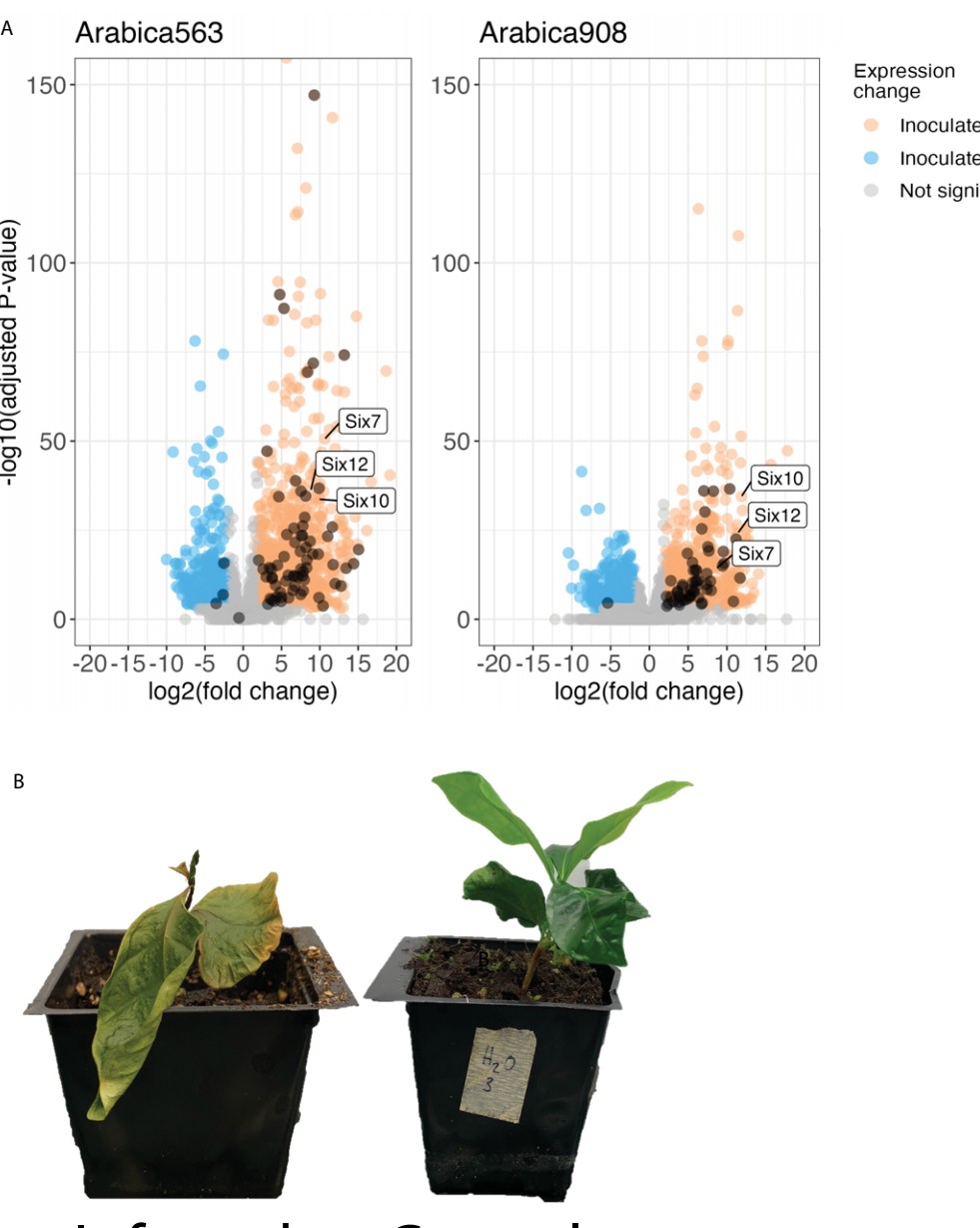

**Fig 5. Up-regulated genes in *Fusarium xylarioides* coffee wilt infection.** (A) Points represent individual genes plotted by log2 fold-change in expression level on the X-axis and significance (–log10 FDR) on the Y-axis. Significantly up-regulated genes in planta samples were identified using negative binomial generalised linear models and shaded orange and those down-regulated are shaded blue. Up-regulation included the *F. oxysporum SIX*, that is secreted in xylem, effectors (annotated), and pectate lyase and other CAZymes involved in pectin metabolism (shaded dark orange). The CAZymes encompass AA9, CE12, CE5, GH105, GH11, GH12, GH43, GH5, GH51, PL1, PL3, PL4, PL9, and PL26. (B) Arabica coffee plants 99 days post inoculation, infected at left, that is inoculated with *F. xylarioides* arabica563, and control at right, that is mock-inoculated with sterile water. The data underlying this figure can be found in https://doi.org/10.5281/zenodo.13836286.

contained the most *mimps*, with *F. oxysporum* f. sp. *lycopersici* and *F. oxysporum* f. sp. *raphani* in the highest copy numbers (90 and 85, respectively, Fig 6). The *mimp* families 1, 2, and 4 were the most numerous in all genomes. Within *F. xylarioides*, robusta and coffea strains

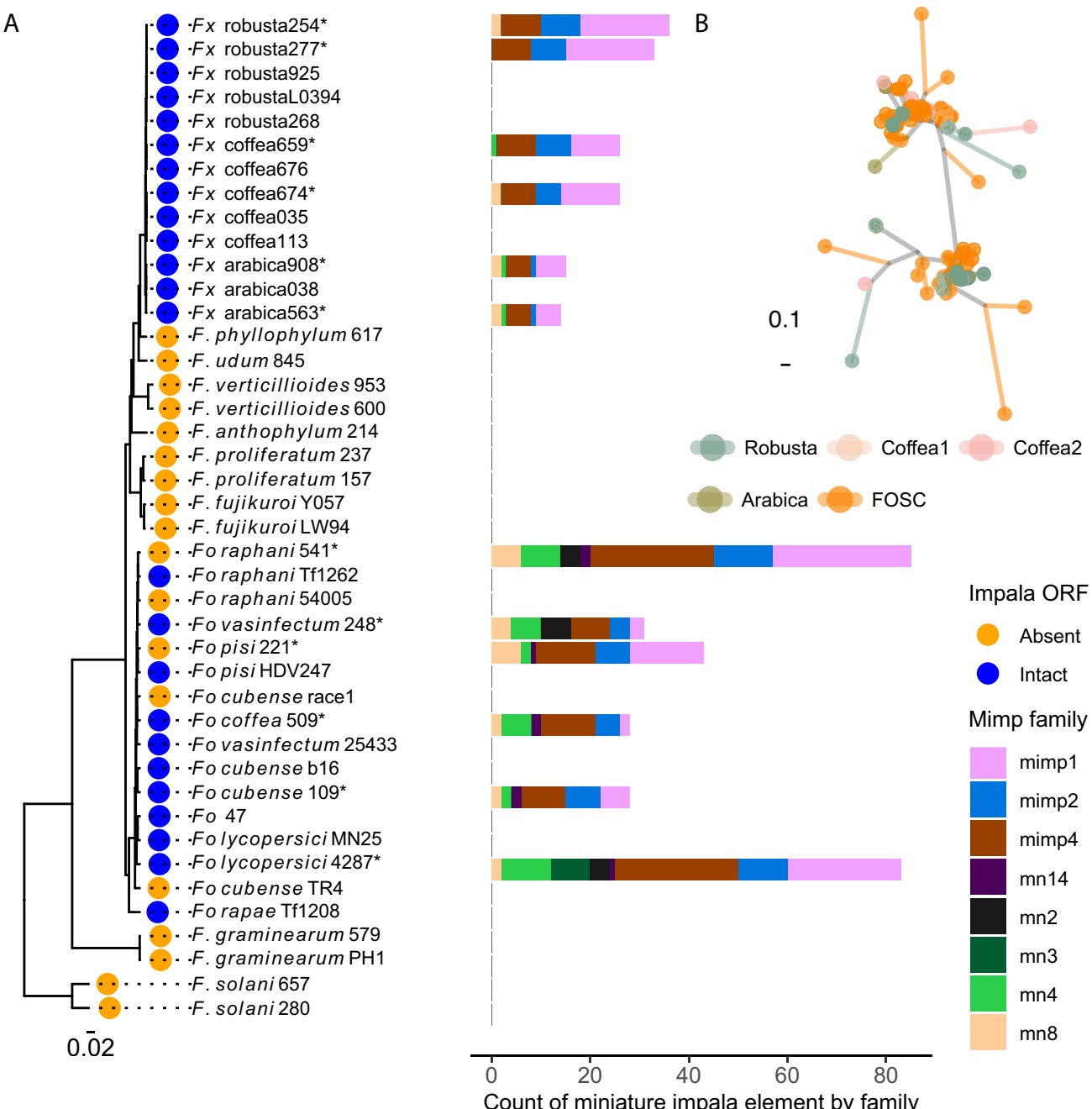

**Fig 6. Miniature impala composition of *Fusarium* genomes.** (A) Miniature *impala (mimp)* elements across the whole-genome are shown as a count per family. They were identified in 6 representative *F. xylarioides* genomes, all *F. oxysporum* formae speciales genomes sequenced in this study and *F. oxysporum* f. sp. *lycopersici* (GCA_000149955.1), each annotated strain denoted by an asterisk. A BLAST search did not return any hits in other genomes shown in the tree. Colour represents the family to which the mimp sequence belongs. The STAG species tree is shown on the left and the tip label corresponds to the presence (blue) or absence (orange) of an intact *impala* transposase open reading frame. The scale bar indicates branch length (substitutions per site). (B) *Fusarium oxysporum* and *F. xylarioides* miniature *impala* (*mimp*) family 1 sequence genealogy. All annotated *mimp* 1 sequences were aligned, with phylogeny represented by an unrooted tree. Each point corresponds to a *mimp* sequence and the scale bar indicates branch length (0.02 substitutions per site). Tree tip labels are shaded by phylogenetic group: arabica, robusta, coffea1, and coffea2 *F. xylarioides* populations; *FOSC, F. oxysporum* species complex. Full strain details in Table 2 and expanded tree in S11 Fig. The data underlying this figure can be found in https://doi.org/10.5281/zenodo.13836286.

contained twice as many *mimps* as the arabica strains, primarily driven by families 1 and 2 (Fig 6). All *F. xylarioides* genomes and most of the *F. oxysporum* genomes contained intact *impala* transposases (Fig 6A), required for *mimp* activity [35]. A small number of *mimps* were found in *F. proliferatum* (also reported in [32]), but no other *mimps* or *impala* transposases were found in the genomes of other species in the *F. fujikuroi* species complex. Genealogy reconstruction showed that *mimps* in *F. xylarioides* are intermingled with those from the *F. oxysporum* species complex, rather than forming separate clades, consistent with either a frequent and ongoing transfer between the 2 species, or a single transfer of multiple diverse copies: multiple clades with strong bootstrap support contain copies from both *F. xylarioides* and *F. oxysporum* (shown for *mimp* family 1, Figs 6B and S11). The alternative hypothesis of ancestral polymorphism would assume complete loss of all these families in all species from the *F. fujikuroi* complex.

Some *mimps* were observed in large regions identified as putative HTRs above. For example, in HTR 2 and 6 that match *F. oxysporum* f. sp. *lycopersici*'s supercontig 51, *F. xylarioides* arabica contained at least 3 and robusta at least 2 *mimps*. Some *mimps* overlapped with- (*six10* and *six12*) or were close to the *six7* effector genes (S7 Fig). The coffea scaffolds that matched the region of supercontig 51 found in *F. xylarioides* arabica also contained *mimps*, while those matching the region found in *F. xylarioides* robusta lacked these. Both robusta and coffea scaffolds contained an unclassified TE ("rnd-6 family-1942") and newly recognised TEs (Fot6, MGR583-like, Yaret1) found on *F. oxysporum* f. sp. *lycopersici*'s mobile pathogenic chromosome (identified by [25]). It is possible, however, that the *mimps* assembled to different coffea scaffolds, as the robusta scaffolds that match supercontig 51 are only 20 kb each.

In addition to *mimps*, all HTRs were high in TEs. Across the *F. xylarioides* arabica563 genome, average TE density (excluding simple and low complexity repeats) was 0.06 TEs/bp, while all HTRs contained more transposons (S11 Table) with an average 0.12 across HTRs. HTRs 2, 5, and 6 had the highest overall TE densities at 0.2, 0.11, and 0.12 TE/bp. HTRs 1–2 and 5–6 all contained diverse TEs which were first identified on the mobile pathogenic chromosome of *F. oxysporum* f. sp. *lycopersici* by [25] (S10 Fig) and the unclassified TE family ("rnd-6 family-1942") which we found with mimps (S7 and S10 Figs and S11 Table). Overall, the forty-four 10-kb windows comprising the HTRs were significantly enriched for *mimps* (total 6 copies) and all classes of TEs known from *F. oxysporum* mobile chromosomes (total 38): probability of observing as many or more in randomly selected forty-four 10-kb windows from the genome $p < 0.001$ (randomisation test, 1,000 trials).

## An 80-kb *Starship* mobile element has inserted between highly expressed genes in the arabica genomes

The role of large mobile elements such as *Starships* in the mobility of effector genes has recently been discovered [15]. A *Starship* mobile element was identified in all 3 arabica genomes on chromosome 11 and fully corresponds to HTR 4, thus we re-define HTR 4 as a *Starship*. This *Starship* was bounded by flanking target site duplications, the 5′ end of the element contained a "captain" gene encoding a predicted tyrosine recombinase with a DUF3435 domain (Pfam accession PF11917) and comprising 22 genes ("cargo") (Fig 7A).

Multiple sources of evidence suggest that this *Starship* is an HTR from *F. oxysporum*. First, this region was independently determined as an HTR with different methods (as HTR 4) and shows high similarity to *F. oxysporum* detected using BLASTn (S6B Fig). In addition, this *Starship*/HTR 4 was found to have high TE density, and specifically TE families which were first identified from the mobile pathogenic chromosome of *F. oxysporum* by Schmidt and colleagues [25] (S10 Fig). Third, the genealogy of the DUF3435 captain gene reveals the *F.*

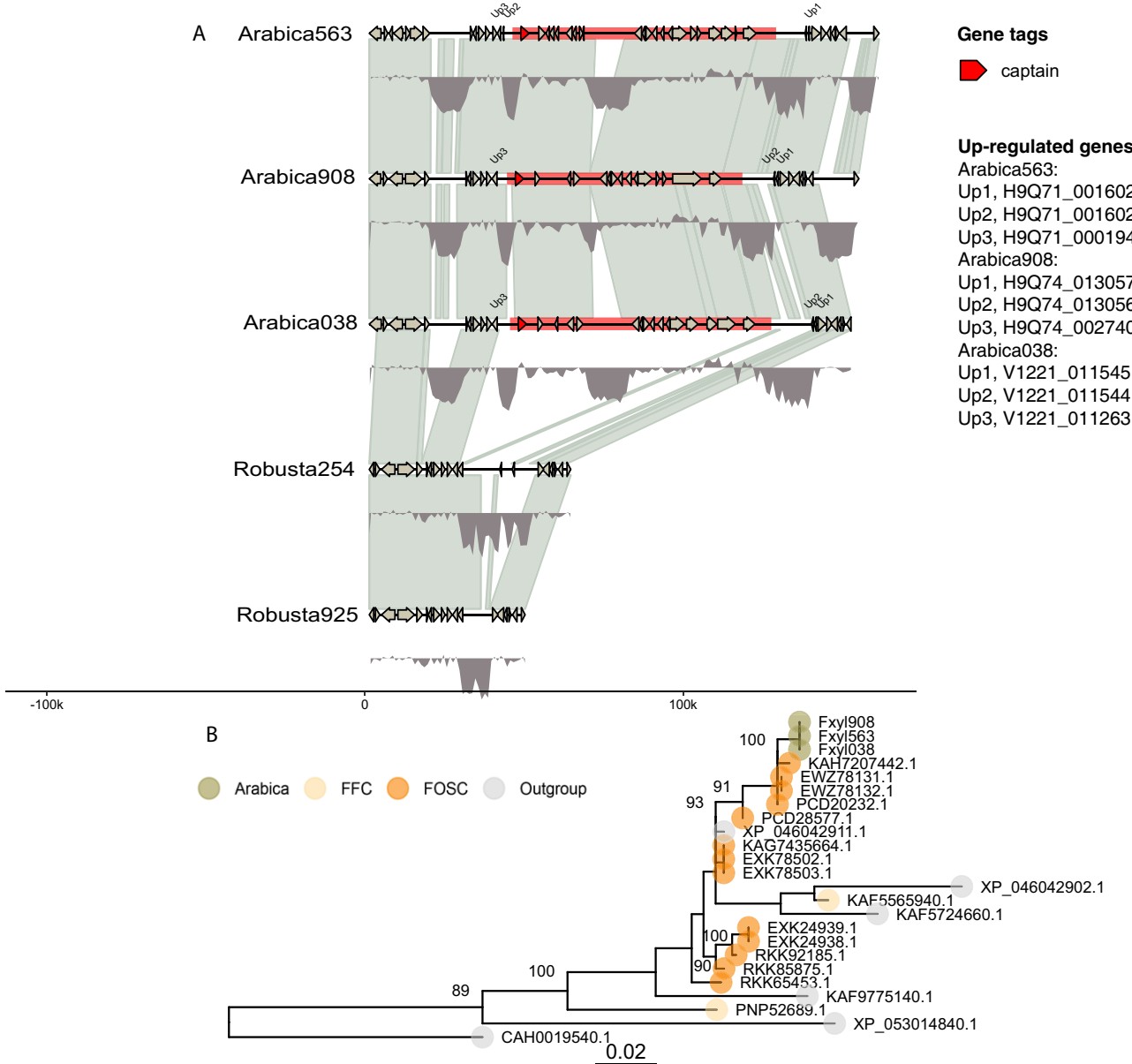

**Fig 7. A *Starship* mobile element in the *Fusarium xylarioides* arabica genomes.** (A) Schematic plot showing known *Starship* tyrosine recombinases (captains), and full-length *Starship* elements coloured red, with predicted gene models shown. GC content is indicated with density plots coloured grey below each genome. Predicted gene models and their direction are indicated, with grey shaded areas indicating synteny. Orthologous gene models are present in each *Starship* (BLAST, E-value $<10^{-150}$); any gaps or differences are due to annotation errors. Up-regulated genes are annotated and named in the key. (B) Maximum likelihood tree showing phylogenetic relationships of *F. xylarioides* tyrosine recombinases searched against the BLAST nonredundant database. Trees are drawn with 1,000 bootstraps and nodes are annotated with bootstrap support. The scale bar indicates branch length (0.02 substitutions per site). The data underlying this figure can be found in https://doi.org/10.5281/zenodo.13836286.

*xylarioides* copies are nested within *F. oxysporum* copies (Fig 7B). The level of divergence between *F. xylarioides* and the nearest *F. oxysporum* copy (*D FxFo* = 2.7%) falls within the lower 0.1% expected across background genes (S14 Table). Interestingly, this *Starship* appears to occupy a TE island (with an AT basis and high repeat density 0.06 TE/bp) and has significantly up-regulated effector-like genes directly flanking either side: H9Q71 0016025 was one

of the top 20% differentially expressed genes and is a nucleartransport factor 2; H9Q71 0016024 is a glycoside hydrolase 43 CAZyme (GH43) predicted by EffectorP as an apoplastic effector; H9Q71 0001940 is a secreted protein not predicted as an effector by EffectorP. Finally, 11 of 12 TEs (>500 bp) within the *Starship* are single copy, i.e., there are no other copies (BLAST, >90% grade) outside the *Starship* in the *F. xylarioides* arabica genomes, except for 1 TE (TE2649, "rnd-6 family-788") which is 12,257 bp and is also found in HTR 5 on supercontig 2. These different lines of evidence point to a recent acquisition of the *Starship* by the common ancestor of *F. xylarioides* arabica genomes.

## Discussion

*Fusarium xylarioides* is genetically differentiated into multiple host-specific populations isolated from different species of coffee. We found that the arabica population shares several highly similar regions with *F. oxysporum*, including a large mobile *Starship* element. These regions are at non-homologous locations in the 2 species and lacking in other closely related species or populations, consistent with their acquisition by horizontal transfer. Transcriptome data following infection revealed high expression of putative effector genes, including some only known to *F. oxysporum* outside of coffee wilt, as well as pectinolytic enzymes. Finally, we found that all *F. xylarioides* populations share highly similar *mimps* and other transposable elements with *F. oxysporum* that have likely transferred between the species and could have facilitated HGT of wider genome regions.

The *F. xylarioides* arabica and robusta groups were genetically differentiated from each other and from coffea1 and coffea2, which were themselves 2 differentiated populations. The 2 coffea populations overlapped spatially and temporally (coffea 1 was isolated from Guinea and Central African Republic, coffea2 was isolated from Guinea, the Central African Republic, and Cote d'Ivoire), yet they remained genetically distinct, with the robusta population apparently emerging from within coffea 1. Possible causes of genetic differentiation could be the occurrence of distinct clonal lineages, consistent with the congruence between gene genealogies within populations, or reproductive isolation due to pleiotropy between host adaptation and mate choice [36]. The specific coffee tree hosts from which *F. xylarioides* coffea population strains were isolated were not recorded, so we do not know if these infected a broad range or specific coffee tree species. Reproductive isolation can be a pleiotropic effect of specialisation if mating occurs within the host [36], as only strains able to infect the same trees can mate. In fact, mating appears to occur in the host in *F. xylarioides* as perithecia are only observed on coffee wilt diseased trees [37], which could have aided genetic differentiation between geographically overlapping populations. Our findings support earlier work based on DNA markers and crossing experiments [38,39], which suggested that *F. xylarioides* was a species complex containing distinct arabica and robusta populations. Therefore, we confirm the presence of genetically differentiated populations with different host specificity. Future taxonomic work including mating tests could determine whether these populations should be named as separate species.

Several lines of evidence support the acquisition of large HTRs into *F. xylarioides* populations over an alternative hypothesis of vertical inheritance followed by loss from other lineages within the *F. fujikuroi* clade. First, a vertical inheritance and loss scenario would require independent losses in at least 6 lineages across the *F. fujikuroi* phylogeny. Second, the regions display significantly higher sequence identity with their nearest *F. oxysporum* match than expected based on background genome levels, whereas vertical inheritance and loss would predict sequence identity to fall within the observed distribution of identities for similarly sized genome regions. Third, HTRs are present in different chromosomal locations in *F. xylarioides*

compared to *F. oxysporum*, in contrast to the high degree of synteny observed across the genome as a whole. Presence in different chromosomal locations is also not consistent with a scenario of introgression at homologous loci following interspecific sexual hybridisation. In theory, horizontal transfers can occur through transposon-assisted mechanisms or by transfer of an entire chromosome [21]. While we have no direct evidence for mechanisms here, either seem plausible. Filamentous fungi, including the genus *Fusarium*, species are known to form anastomoses (somatic fusions between mycelia) which may facilitate the transfer of genetic material by whole chromosomes or potentially by other mobile elements [40]. Indeed, hyphal fusions have been shown to occur within the arabica and robusta *F. xylarioides* populations [41]. The finding of specific shared transposons between *F. xylarioides* and *F. oxysporum*, and in particular a *Starship* element, known in other species to be able to transfer horizontally [15], would be consistent with a role for mobile elements in horizontal transfer.

Each of the HTRs appears to represent a single transfer event into the most recent common ancestor of the *F. xylarioides* population (or populations for HTR 2 and 5) from *F. oxysporum*: locations of HTRs are conserved within each population, and there is low sequence variation compared to largescale variation observed among *F. oxysporum* copies. Transfer of pathogenicity genes via accessory chromosomes was first reported in *F. oxysporum* f. sp. *lycopersici*, with horizontal transfer of its entire mobile pathogenic chromosome to 2 previously non-pathogenic strains [21]. Several additional cases have since been reported, each with an association with *mimps* in transposition [25,32]. In *F. xylarioides*, a whole mobile chromosome may have been acquired by a common ancestor to the arabica and robusta populations, with later divergence and differential loss of genomic regions and integration of different parts. Loss of regions that are no longer useful to the organism is well reported [42]. Alternatively, only parts of the *F. oxysporum* chromosome may have been transferred into *F. xylarioides*, with different parts in arabica and robusta populations.

The association of *mimps* with the HTRs sheds further light on how transposable elements might interact with host specialisation. Overall, we found 4 putative HTRs with TEs that have been linked to horizontal transfer and the generation of effector variability in *F. oxysporum*. TEs could generate effector variation either via direct transposition (for example, [43] and as in HTRs 1–3 and 5) or via increasing the potential for recombination events (including ectopic recombination and genome loss). One HTR, the *Starship*/HTR 4, lacked *mimps* and the transposons identified from the *F. oxysporum* mobile pathogenic chromosome, but contained other single-copy TEs. We found few *mimps* in the *F. fujikuroi* complex species (see also [32]), whereas arabica and robusta *F. xylarioides* populations contained different numbers of *mimps* in total as well as in their HTRs, consistent with different histories of transfer and/or expansion in each population. In *F. oxysporum*, *six10*, *six12*, and *six7* make up supercontig 51 and are flanked by *mimps* [25]. In *F. xylarioides*, different pieces of supercontig 51 are present in the arabica and coffea genomes, versus the robusta and coffea genomes. The *SIX* genes in arabica were flanked by *mimps*, while the robusta region also contained *mimps* and other class II TEs, as well as other genes involved in pathogenicity. A study of 2 *F. oxysporum* f. sp. *cubense* transcriptomes found that both lacked mobile chromosomes of *F. oxysporum* f. sp. *lycopersici*, yet still showed differing presence and expression of *SIX* genes [44]. In *F. oxysporum*, all effector genes contain a *mimp* in the promoter region [25], yet in *F. xylarioides*, this was only the case for the 3 *SIX* genes. In addition, the fact that the *mimp*-overlapping *SIX* genes are only found in the arabica and coffea populations (and not robusta nor any *F. fujikuroi* complex species) suggests a horizontal transfer origin.

To our knowledge, such patterns of horizontal transfer of pathogenicity between species complexes has not been reported in *Fusarium* before, while introgression has [45]. We found that the closest match to one of our putative HTRs (HTR 4/*Starship*) was *F. oxysporum* 509,

sequenced in this study and isolated from an arabica coffee plant in the 1970s in Tanzania, which is spatially and temporally within the robusta coffee wilt disease range. This suggests these patterns of horizontal transfer could be more commonplace than thought. The close phylogenetic relationship between CAZyme gene families from *F. oxysporum* and the arabica strains provides further evidence of horizontal transfer of gene types that are important in vascular wilt infection. Such horizontal transfer could have occurred in a shared niche, as both *F. xylarioides* and *F. oxysporum* are soil-borne pathogens and have been isolated from the roots and wood of coffee wilt-diseased trees in Ethiopia and central Africa [46], as well as from banana roots in an intercropped field in Uganda [47], and on which *F. oxysporum* f. sp. *cubense* is widespread [48]. These transfers may have driven the evolution of new host-specific populations and therefore could have facilitated disease emergence and subsequent re-emergence.

Recent comparative genomics studies have elucidated the role of these changes in domesticated fungi and plant pathogens, including a host jump in *Phytophthora infestans* followed by adaptive specialisation [49], transfer of a 14 kb *Starship* encoding the ToxA virulence protein and abundant transposons between 3 wheat fungal pathogens [15], *Starship* activity in the human pathogen *Aspergillus fumigatus* which promotes strain heterogeneity [18], and the transfer of regions that allow adaptation to cheese environments in distantly related species [50]. Furthermore, transfer of pathogenicity into a previously nonpathogenic strain in *F. oxysporum* has been demonstrated experimentally [21]. These examples, as well as cross-kingdom transfers (e.g., [51]), demonstrate the occurrence of HTRs across many distantly related branches of the eukaryotic tree of life and their importance for shaping eukaryotic evolution [52]. Previously, the analysis of historic genetic material involved the use of highly degraded museum specimens and molecular markers and target probes in a small number of genes [53]. However, fungal culture collections contain tens of thousands of fungal isolates collected over the past century and preserved in a living state [54]. Connecting collections with new-omics studies provides a powerful way to investigate the role of HTRs in historic fungal outbreaks.

## Materials and methods

### Strain details

All strains were from the CABI culture collection (Egham, United Kingdom), with IMI 507113 transferred from the Belgian coordinated collection of microorganisms from the Université Catholique de Louvain (Louvain-la-Neuve, Belgium) (under MUCL 47066/ MNHN 709) and IMI 507035 and 507038 transferred from the Agricultural Research Service culture collection (Illinois, United States of America) (with 507035 under NRRL 25804/BBA 62721/CBS 749.49; and 507038 under NRRL 37019/FRC L96/BBA 62458). The 11 additional *Fusarium* strains were selected for sequencing based on the presence of certain putative effector genes described in [28], which was established by Sanger sequences of PCR data [55].

### Long-read reference genome sequencing and assembly

The *F. xylarioides* strain IMI 389563 (hereafter called "arabica563") from the arabica host-specific strain group of coffee wilt disease was used for long-read sequencing, and is available from the CABI-IMI culture collection (CABI, Egham, UK). Arabica563 mycelium was grown in GYM broth for 7 days at 25°C and 175 RPM. High-quality high molecular weight genomic DNA was extracted using the chloroform extraction method in [56]. Specifically, 1 to 2 g of mycelium were ground in liquid nitrogen using a lysis buffer containing sodium metabisulfite, 2 M Tris, 500 mM EDTA, 5 M NaCl, and 5% Sarcosyl. The gDNA was extracted from the cell lysate using chloroform, and precipitated using isopropanol and 70% ethanol. The gDNA was

eluted in 500 μl TE (Tris EDTA) buffer and incubated overnight at 4°C. RNA contaminants were removed using RNAse A (10 mg/ml) and gDNA was purified using Agencourt AMPure XP beads (Thermo Fisher). The gDNA quantity was determined using a NanoDrop 2000 (Thermo Fisher) and its integrity was determined by electrophoresis with a 0.5% agarose gel. gDNA (1 μg) was diluted in nuclease-free water to a final volume of 47 μl. Half of the DNA was then sheared into 20 kb fragments with a COVARIS g-TUBE (Covaris, Woburn, Massachusetts, USA), centrifuging at 4,200 rpm for 1 min to increase DNA throughput. The sheared DNA was converted into Oxford Nanopore libraries using the SQK-LSK110 library kit and the short fragment buffer (Oxford Nanopore Technologies, United Kingdom). Library prep followed the standard manufacturer's protocol apart from the final elution, which was done for 30 min at 37°C. The libraries were sequenced using a SpotON R9.4.1 FLO-MIN106 flowcell for 48 h.

Nucleotide bases were called from the raw sequence data using the super high accuracy model (SUP) of the GPU version of Guppy v6.0.1 (config file: dna r9.4.1 450bps sup.cfg). Base-called reads were assembled into contigs by Flye v2.9 with "-nano-hq" parameters [57]. The Flye assembly was error-corrected using both Nanopore and Illumina data with 1 round of medaka v1.4.4 (https://github.com/nanoporetech/medaka) and 1 round of (Hapo-G) polishing [58]. Contig completeness was visualised in Tapestry v1.0.0 [59], including the presence of the terminal telomere repeat sequence TTAGGG, common to eukaryotes. The presence of core *Fusarium* chromosmes, shared by close and distantly related sister species was assessed by mapping the *F. xylarioides* arabica563 reference to the *F. verticillioides* chromosomal assembly. First, the 2 genomes were aligned using nucmer [60], before visualising the alignment in Dot-Prep.py (https://github.com/marianattestad/dot). Those contigs in *F. xylarioides* which match to multiple *F. verticillioides* chromosomes are described as supercontig 1 and supercontig 2.

## Illumina sequencing and assembly

An additional 11 *Fusarium* genomes were selected for Illumina whole-genome sequencing: 5 *F. xylarioides*, 5 *F. oxysporum*, and 1 *F. solani* (Table 1), see Strain details. Of the 5 *F. xylarioides* strains, 3 were from the pre-1970s outbreak, which was observed to infect multiple *Coffea* species and which we call "coffea" (coffea035, coffea113, coffea676), and the remaining 2 were the earliest isolates available in the collections for each of the arabica and robusta groups (both isolated from around 1970, arabica038 and robusta268).

Strain morphologies were verified and grown in GYM broth following the same protocol as above. DNA was extracted from 250 to 500 mg of washed mycelium following the NucleoSpin Soil Mini Kit (Macherey Nagel, Germany) standard protocol. For each strain, a single library was prepared with the Illumina DNA PCR-Free Prep and sequenced with the Illumina Nova-seq 6000 with 2× 150 bp reads. Low-quality bases (Phred score <20) and adapters (stringency 4) were removed using TrimGalore 0.6.0 by Cutadapt (https://github.com/FelixKrueger/TrimGalore, [61]). Overlapping paired-end reads were stitched together using FLASH 1.2.11 [62] and de novo assembled with MEGAHIT 1.2.9 [63,64]. Assembly metrics including BUSCO scores were computed using QUAST 5.2.0 (Table 1) [65].

Each *F. xylarioides* MEGAHIT assembly from this study, as well as those assembled previously [28] and the publicly available genomes robusta925 [66] and robustaL0394 [67], was mapped to the reference genome assembly using RagTag v2.1.0 homology-based scaffolding [68]. The parameter "-u" identified unmapped (and therefore absent from arabica563) scaffolds. The *F. oxysporum* and *F. solani* MEGAHIT assemblies were mapped to their closest and least fragmented publicly available assembly (Table 2). These RagTag assemblies were used to predict TEs (see later section).

The raw sequence data and assembled genomes have been deposited in the International Nucleotide Sequence Database Collaboration (INSDC) database under the BioProject accession number PRJNA1043203.

### RNA sequencing to aid annotation of long- and short-read assemblies

The same arabica563 strain used for long-read sequencing was grown under the same culture conditions described above. Mycelia were harvested and total RNA extracted using the standard protocol of the RNeasy Mini kit (Qiagen, Hilden, Germany). The RNA quality assessment (Qubit and Fragment Analyzer), library preparation and Next Generation Sequencing (60 M paired-end reads per sample) were performed by Genewiz/Azenta (Frankfurt, Germany). See RNA extraction section for QC steps. Transcript sequences were de novo assembled using Trinity v2.13.2 with the genome-free method [69].

### Gene prediction and annotation

Protein-encoding genes were predicted in all short-read genomes using the Funannotate Eukaryotic Genome Annotation Pipeline v1.8.11 [70]. All 13 *F. xylarioides* genomes were annotated (Table 1), including re-annotations for the 6 genomes from [28]. Before annotation, repetitive contigs were removed using "funannotate clean" with default parameters and repetitive elements within the assemblies were soft masked using RepeatModeler and RepeatMasker with "funannotate mask" and *F. oxysporum* parameters.

The trimmed RNA-seq reads and the Trinity assembly were aligned to the arabica563 MEGAHIT assembly using the parameters "funannotate train --jaccard_clip" which runs PASA to model the gene structures. The PASA gene models were parsed to run Augustus, snap, GeneMark, and GlimmerHMM, with the predictions used to run CodingQuarry and the Funannotate Evidence Modeler. The complete set of arabica563 output files were then used to predict protein encoding genes on the remaining MEGAHIT genome assemblies using "funannotate predict -p predict results/fusarium xylarioides 389563.parameters.json." The arabica563 MEGAHIT assembly was selected over the reference genome to avoid issues with gene prediction on assemblies of different quality when the parameters were applied to the remaining *F. xylarioides* MEGAHIT assemblies. The gene model predictions were refined (e.g., correcting intron-exon boundaries) and untranscribed regions annotated by re-aligning the RNA-seq data for each genome using "funannotate update" on the output files from "funannotate predict." The annotations were then mapped to the arabica563 reference genome and long-read RagTag assemblies using Liftoff [71] with default parameters.

### Other *Fusarium* genomes

The same steps were followed for the 5 *F. oxysporum* genomes and the *F. solani* genome, using different sources of evidence for the different species. RNA-Seq reads for the closest *formae speciales* match were downloaded from the sequence read archive (see Table 1 for accession numbers) and aligned to the corresponding assembly using "bbmap.sh"), and expressed sequence tag data for the closest *f. sp.* match from JGI Mycocosm were used to provide evidence to "funannotate predict." The expressed sequence tag data were produced by the US Department of Energy Joint Genome Institute (https://www.jgi.doe.gov/) in collaboration with the user community. Evidence sources are listed in Table 2. The same RNA-Seq data was then re-aligned through "funannotate update" as in the section above.

Functional annotation of the protein-coding genes for all genomes was completed with annotations from PFAM, InterPro, UniProtKB, MEROPS, CAZyme, and GO Ontology through "funannotate annotate." InterProScan v5.56–89.0 was run locally using the parameters

**Table 2. Data sources used to annotate *Fusarium oxysporum* and *F. solani* genomes sequenced in this study.** The columns describe: species, CABI IMI strain number, RNA-seq data used as evidence in gene prediction, expressed sequence tag (EST) data used as evidence in gene prediction, which publicly available assembly the genome was scaffolded to using RagTag for annotation of repeats. Strain bio-geographic details are in S1 Table.

| Species | Strain number (IMI) | RNA-seq data | Expressed sequence tag data | Genome assembly for scaffolding |
|---|---|---|---|---|
| *F. oxysporum* f. sp. *cubense* | 141109 | ERR10015931 | *F. oxysporum* f.sp. *cubense* II5 v2.0 [72] | GCA_005930515.1 |
| *F. oxysporum* f. sp. *coffea* | 244509 | SRR10123445 | *F. oxysporum* f.sp. *cubense* II5 v2.0 [73] | GCA_000149955.1 |
| *F. oxysporum* f. sp. *vasinfectum* | 292248 | SRR12855423 | *F. oxysporum* f.sp. *cubense* II5 v2.0 [73] | GCA_000260175.2 |
| *F. oxysporum* f. sp. *raphani* | 337541 | SRR14318405 | *F. oxysporum* f.sp. *cubense* II5 v2.0 [73] | GCA_019157275.1 |
| *F. oxysporum* f. sp. *pisi* | 500221 | SRR12855423 | *F. oxysporum* f. sp. *pisi* F23 v1.0 | GCA_000260075.2 |
| *F. solani* | 392280 | SRR19182467 | *F. solani* FFSC 5 v1.0 [73] | GCA_020744495.1 |

"-goterms -iprlookup" and phobius was run through the web browser (https://phobius.sbc.su.se/). All proteins lacking significant hits were annotated as hypothetical proteins.

### Transposable elements and repeat annotations

Repeatss and transposable elements were identified from the nucleotide assemblies. A custom repeat library was constructed using all *Fusarium* RagTag assemblies in RepeatModeler v2.0.3 with the parameter "-LTRStruct" [74]. Added to this library were TEs known to be involved in *F. oxysporum* transfer of pathogenicity between strains: *impalas*, a new set of TEs described by [25], as well as manually curated *mimps*. Sequences for *mimp* families 1–4 were downloaded from NCBI (accession numbers: AF076624.1, AF076625.1, EU833100.1, and EU833101.1). Sequences for the more divergent *mimp* families mimp5, mimp6, mn2, mn3, mn4, mn8, and mn14 are from [35]. Manual curation followed the protocol described by [75] and involved: identifying conserved domains with sequence homology to these TEs using BLASTn; only retaining the best BLAST hits ($\geq$ 50% query length and $\geq$ 80% identity) for each TE family; extracting and aligning nucleotide sequences for TE families including an additional 500 bp up- and down-stream; trimming alignments to the TIRs at their start and end and removing insertions using T-COFFEE [76]. This generated consensus sequences for each *mimp* family that were added to the RepeatModeler custom library and used to call TEs and repeats from genome scaffolds using RepeatMasker v4.1.2 with the parameters "-lib $LIB -gff -no_is" [77]. The presence of the *impala* transposase ORF from *F. oxysporum* f. sp. *melonis* was found by a BLAST search "tblastn -evalue 1e$^{-5}$" using accession AAB33090.2. See Data Availability for custom repeat library and analysis scripts.

The *mimp* family 1 phylogenetic tree was constructed by aligning all *mimp* family 1 sequences that were annotated with MAFFT v7.309 [78,79] in Geneious v9.1.8 (https://www.geneious.com). We used Geneious to select the resulting consensus sequence and used BLASTn [80] in Geneious to search for copies in the public genomes used in this study (S2) as well as the NCBI "nr" database. Copies were only selected if they were >80% in length and identity and were a species other than *F. xylarioides* and *F. oxysporum*: just 1 additional species was included (*F proliferatum*). Sequences were re-aligned using MAFFT in Geneious, with maximum likelihood trees constructed in IQ-Tree2 v2.3.5 with 1,000 bootstrap replicates, with the best-fit model automatically selected by ModelFinder and using the ultra-fast bootstrap parameter [81]. Phylogenetic relationships were visualised with ggtree.

## Orthologous gene groups, species tree, and genetic structure

Whole-genome similarity between the protein-encoding genes was assessed using NSimScan, part of the QSimScan package [82]. OrthoFinder v2.5.4 was used to determine *Fusarium* orthologous groups between the genomes sequenced in this study (Table 1) and published genomes detailed in S2 Table. OrthoFinder infers gene trees for all orthogroups and uses gene duplication events via the STRIDE algorithm to root them [83]. Specifically, unrooted gene trees are built for each orthogroup using DendroBLAST [84] and trees are resolved using the OrthoFinder hybrid species-overlap/duplication-loss coalescent model [85]. The "supergene" species tree was built and rooted using STAG in OrthoFinder [86,87] with 3,544 orthogroups in which every strain has at least 1 gene copy. Resolved monophyletic species clades were identified using tr2 software [88]. An alternative "supermatrix" species tree was reconstructed by concatenated alignment of all 1,685 single-gene orthologs using seqkit [89], creation of a partition file and maximum likelihood analysis using IQ-Tree2 v2.3.5, the partition file, the best-fit model automatically selected by ModelFinder and the ultra-fast bootstrap [81].

SNPs were called against the arabica563 reference genome for each set of *F. xylarioides* reads, with reads for robusta925 downloaded from the Sequence Read Archive (accession PRJNA508603, [66]) and robustaL0394 shared by [67]. SNPs were called using GATK v4.1.2.0 "HaplotypeCaller -ERC GVCF," which provides 1 genomic variant call format (gVCF) output file per strain. Using Snakemake v5.3.0, gVCFs were combined using GATK "CombineGVCFs," genotypes were combined with "GATKGenotypeGVCFs," "SelectVariants -selecttype SNP" selected the SNPs which were then filtered with "Variant-Filtration QUAL<30, DP<10, QD<2.0, FS>60.0, MQ<40.0, SOR>3.0, QRankSum<-12.5, ReadPosRankSum<-8.0." The SNP data set was used in Neighbour-Net analysis using SplitsTree App (version 6.3.37, [90]) to infer population structure. Absolute nucleotide divergence $d_{xy}$ was calculated using the R package Popgenome [91].

## Identification of *Fusarium xylarioides* population-specific regions and horizontal transfers

*Fusarium xylarioides* population-specific regions were identified by mapping each *F. xylarioides* genome to the arabica563 reference using minimap2 with the parameters "-cx asm10 --secondary=no --cs" [92]. The PAF file was converted to bed using "paftools.js splice2bed," and then coverage was calculated in 100-bp windows using "bedtools coverage." Finally, coverage was calculated in contiguous 10-kb windows by taking the average of $100 \times 100$ bp windows using "bedtools map -c 7 -o mean." Mean mapped coverage was calculated for each population group (arabica, robusta, coffea1, and coffea2) for each 10-kb window. Regions were identified as: arabica-specific if they were present (mean similarity >0.5) in arabica and absent (mean similarity = 0) from all other population groups (48 windows); arabica- and coffea1-specific if the arabica and coffea1 means >0.5 and the coffea2 and robusta means <0.2 (22 windows); arabica- and coffea2-specific if the arabica and coffea2 means >0.5 and the coffea1 and robusta means <0.2 (21 windows); arabica-, coffea1-, and coffea2-specific if these means were >0.5 and robusta was <0.2 (9 windows); and arabica- and robusta-specific if arabica and robusta means were >0.5 and coffea1 and coffea2 were <0.2 (15 windows). See Data Availability for analysis scripts and data files.

Putative horizontal transfers were identified using BLASTn [80] in Geneious v9.1.8 (https://www.geneious.co) to search for these regions in their entirety in the NCBI "nr" database. Only those hits with a grade >50% (grade combines query coverage, *E*-value and identity values for each hit with weights 0.5, 0.25, and 0.25, respectively; all *E*-values >1e<sup>-133</sup>) were selected, namely the longest, highest identity hits. We describe as putative HTRs where such hits were

only present in, or most closely related between, *F. xylarioides* and *F. oxysporum*. Selected BLAST hits for each putative HTR were aligned using MAFFT v7.309 [78,79] in Geneious and phylogenetic trees reconstructed for each alignment using IQ-Tree2 v2.3.5 with 1,000 replicates of ultra-fast bootstrap branch support [81,93]. This resulted in 5 putative HTRs identified in the *F. xylarioides* arabica563 reference genome on chromosomes 1, 5, 8, 11, and supercontig 2. After discovering that HTR 2 matched a chromosomal subregion "supercontig 51," a region on the *F. oxysporum* mobile pathogenic chromosome which is enriched in effector genes, a BLAST match of each *F. oxysporum* chromosomal subregion against the *F. xylarioides* genomes returned an additional HTR in the robusta genomes (HTR 6). Synteny between the *F. xylarioides* putative HTRs and accessory regions in 2 long-read *F. oxysporum* genomes (*F. oxysporum* f. sp. *lycopersici* and *F. oxysporum* f. sp. *cubense* were visualised by first aligning each to the arabica563 reference using MUMmer's nucmer [60], then plotting the alignment in Rideogram [94]). The remaining and non-horizontally transferred *F. xylarioides* population-specific regions were found across the *F. fujikuroi*, *F. oxysporum*, *F. graminearum*, and other *Fusarium* species complexes. See Data Availability for analysis scripts and data files. The putative HTR bed file and nucleotide sequences are available as S1 and S2 Figs.

*Starships* were identified using starfish v1.0.0 [95] in conjunction with metaeuk [96], hmmer [97], bedtools [98], blastn [80], mummer4 [60], cnef [99], sourmash [100], mcl [101], and visualised using gggenomes [102], mafft [78], and minimap2 [92]. In total, starfish identified 15 putative elements over 19 genomic regions. By manually inspecting each, we determined that one is a *Starship* and the remainder dead and/or degraded. We used BLASTp on NCBI (https://blast.ncbi.nlm.nih.gov/) to search for matching hits to the 3 identical *F. xylarioides* DUF3435 "captain" genes, and aligned using MAFFT in Geneious those hits with a score >1,300 (score is the sum of alignment scores of all segments from the same subject sequence, comprising query coverage and *E*-value). The phylogenetic tree was drawn using PHYML in Geneious with bootstrap branch support calculated with 100 replicates. The *Starship* comprises 47 simple and low complexity repeats, and TEs. We took TEs with length >500 bp (*n* = 12) and used BLAST in Geneious to look for other copies in the *F. xylarioides* genomes. TEs were counted as present with a >90% grade match to the *F. xylarioides* arabica563 copies in the *Starship*. The predicted gene annotations and functions within and flanking the *Starship* are described in S3 File.

## Statistical evaluation of putative horizontal transfers

For each putative HTR, we calculated the average sequence identity (across 10 kb windows) to its closest match in *F. oxysporum* (defined as *I_Fo*) and *F. xylarioides* (excluding *F. xylarioides* strains in which the HTR was determined to be present, *I_Fx*). We then used *I_Fx - I_Fo* as a metric of the identity to *F. oxysporum* relative to the other *F. xylarioides* populations: a negative value indicates greater similarity to *F. oxysporum*. For each HTR, we then calculated an equivalent metric for all similarly sized regions across the genome as a whole, and took the proportion of regions showing a value ≤ that observed for each HTR as the chance of observing the HTR value, given the empirical background distribution. For each HTR and for the captain gene of the *Starship* in HTR 4, we ran an analogous phylogenetic test. First, we calculated the mean patristic pairwise divergence between *F. xylarioides* sequences and their closest *F. oxysporum* match (*D_FxFo*). Then, we calculated the same metric across all single-copy ortholog gene trees as a sample of expected background levels of divergence. The percentile of the observed value for each HTR (and the captain gene) was then used as an estimate of the chance of observing the same pairwise genetic divergence to *F. oxysporum* across the genome.

The putative HTRs were associated with *mimps* and other TEs using a randomisation trial. There are forty-four 10-kb windows which comprise the putative HTRs, which contain mimps

(total 6 copies) and all classes of TEs known from *F. oxysporum* mobile chromosomes (total 38 copies). We randomly selected the same number of windows from the *F. xylarioides* arabica563 genome and counted how many times they contained more copies of *mimps* or other TE classes. We repeated this 1,000 times; there were no instances where the randomly selected windows contained more *mimps* or other TEs than the putative HTRs ($p < 0.001$).

## Infection assays and controls

The *F. xylarioides* strains IMI 389563 and IMI 375908 ("arabica563" and "arabica908") from the arabica host-specific coffee wilt disease population from the CABI-IMI culture collection (CABI, Egham, UK) were used. Strains were grown on synthetic low nutrient agar at 25˚C, with the spores verified after 5 days using lactophenol cotton blue staining under a compound microscope following [103]. The cultures were harvested with 1 drop of Tween20 and in 10 ml sterile water. Two plates were combined to make each fungal inoculum before conidial concentrations were measured with a haemocytometer and adjusted to $10^6$ spores/ml. A third control inoculum was made in the same way using sterile distilled water added to fresh synthetic low nutrient agar that had not been inoculated. Each inoculum type had 4 technical replicates.

*Coffea arabica* plants were purchased online from Gardeners Dream (https://www.gardenersdream.co.uk/) and left for 12 weeks to acclimate and grow in a controlled growth room at 25˚C with a 12:12 hour light:dark cycle with 120 µmol/m$^2$/s wavelength and 50%:65% day:night humidity.

We compared gene expression of *F. xylarioides* strains infecting coffee plants and grown in axenic culture, as well as between the inoculated and control-inoculated coffee plants. Plants were infected with 10 µm through stem wounding [39], using a sterile needle at the first green (i.e., non-woody) internode. Each inoculum type (arabica563, arabica908, control inoculum) was used to inoculate 4 plants, making up 16 replicates in total per inoculum type. Plants were grouped by inoculum type and sealed in propagator trays with parafilm to make 4 trays containing 4 plants for each inoculum type. Tray location in the growth chamber was randomised and the plants were grown under shade at 25˚C with a 12:12 hour light:dark cycle in a growth room until at least 50% of infected plants showed wilting, yellowing or early leaf senescence as symptoms of infection. The plants were harvested 98 days after inoculation.

Axenic cultures of arabica563 and arabica908 isolates were grown in GYM broth at 25˚C with shaking at 175 RPM for 7 days. As with the in planta inocula, the in axenic culture samples included 4 biological replicates (arabica563 1–4 and arabica908 1–4).

## RNA extraction, sequencing, and quality control

All aboveground plant material was collected and flash-frozen in liquid nitrogen. The biological replicates were ground separately in liquid nitrogen with PVPP (polyvinyl polypyrrolidone) in a mortar and pestle. Approximately 5 g of each biological replicate was pooled in a 1.5 ml polypropylene tube, precooled in liquid nitrogen, before vortexing and re-freezing.

Total RNA was extracted using the standard protocol of the RNeasy Mini kit (Qiagen, Germany) and following the same methods described above. In total, 20 samples were extracted with 4 technical replicates for each treatment: in planta arabica563-infected coffee plants; in vitro culture arabica563; in planta arabica908-infected coffee plants; in vitro culture arabica908; control (mock-inoculated with water) coffee plants. RNA was quantified using a Nano-Drop 2000 (Thermo Fisher), with the RNA quality assessment (Qubit and Fragment Analyzer), library preparation and Next Generation Sequencing (60 M paired-end reads per sample) performed by Genewiz/Azenta (Frankfurt, Germany). Raw sequenced reads were quality- and adapter trimmed using Trimmomatic v0.39 (parameters: ILLUMINACLIP:

$TRIMMOMATIC DIR/adapters/TruSeq3PE.fa:2:30:10 SLIDINGWINDOW:4:5 LEADING:5 TRAILING:5 MINLEN:25) [104]. Reads were quality checked with FASTQC v0.11.2 [105] and retained where both pairs were trimmed. To remove ribosomal RNA sequences, reads were mapped to the SILVA rRNA database using BBTools (sourceforge.net/projects/bbmap/— Bushnell B.) "bbmap" with the parameter "outu = filtered R*.fq.gz" to keep only the unmapped reads. Finally, the reads were repaired using BBTools "bbmap" "repair.sh" to re-pair reads that became disordered or had their pair eliminated. These QC steps removed 61 GB of data, see S4 File. Before using the data to quantify gene expression, reads were mapped using "bbmap.sh" to the gene annotations of arabica563 and arabica908 and from *C. arabica* (GCA_003713225.1, [106]). These mapped read libraries were used for all next steps.

## Differential expression analysis

Transcript quantification was performed three times, using the Funannotate gene annotations as the target transcriptomes in a genome-free way using Salmon v1.5.2 [107] on the mapped read libraries: in vitro culture arabica563 and in planta arabica563; in vitro culture arabica908 and in planta arabica908; infected *C. arabica* plants and control-inoculated *C. arabica* plants. The relationships within and between biological replicates for each data set were visually examined using the "PtR" script in the Trinity toolkit [69,108]. Each transcript count matrix was tested for differential expression using DESeq2 [109], which uses negative binomial generalised linear models to test for statistical significance. Differential expression was also calculated using edgeR [110], which gave similar results, and limma/voom [111] which handles biases in RNA-seq data differently [112] and was too conservative with the read numbers from the in planta samples for this experiment. The Benjamini–Hochberg method [113] was used to adjust *P*-values for multiple testing to control the false discovery rate (FDR) and strict significance thresholds were applied, with *P*-values <0.001 and >4-fold to define a set of differentially expressed genes. The top 20% differentially expressed genes were classified as the "most differentially expressed": 272 genes in arabica563; and 177 in arabica908.

## Functional annotation analysis

**Gene Ontology and InterProScan classifications.** Arabica coffee gene annotations (accession: GCF 003713225.1) were assigned InterProScan and Gene Ontology classifications using "interproscan.sh -iprlookup–goterms."

**EffectorP.** A new set of putative effector proteins were identified using EffectorP 3.0 [34] on secreted proteins which were significantly differentially expressed in both fungal strains (total differentially expressed secreted proteins: 224 in arabica908 and 380 in arabica563). Only secreted proteins significantly up-regulated in both strains were described as effector-like.

**Carbohydrate-active enzymes.** Carbohydrate Active EnZymes (CAZy) enzymes are grouped by family in the CAZy database [114]. CAZyme-encoding differentially expressed genes were identified in 2 ways, by comparing the proportion of differentially expressed genes to the genome and taking those where >50% of the genes were significantly differentially expressed in both strains, and by identifying enriched gene ontology terms with a cell-wall or pectin-degrading enzymatic function.

**Data analysis and presentation.** All analyses were performed in R using R Statistical Software (v4.2.1; R Core Team) [115] and in Linux run on the Imperial College Research Computing Service HPC facility (DOI: 10.14469/hpc/2232).

The following R packages were used in analyses and to make figures, all using the latest versions: tidyverse [116], cowplot [117], ggrepel [118], wes anderson [119], pheatmap [120],

ggtree [121], treeio [122], aplot (https://cran.r-project.org/package=aplot - Yu G.), RIdeogram [94], circlize [123], ape [124], viridis [125], pals [126], pafr [127], and gggenomes [102].

## Supporting information

**S1 Fig. The *Fusarium xylarioides* and *F. phyllophylum* clade of the supermatrix species tree based on 1,685 concatenated single-copy orthogroups.** Annotated branches indicate bootstrap branch support values. The scale bar indicates branch length (0.001 substitutions per site). *Fx*, *F. xylarioides*. Full strain details in S1 and S2 Tables. The data underlying this figure can be found in https://doi.org/10.5281/zenodo.13836286.
(TIFF)

**S2 Fig. STAG supergene species tree based on 3,544 orthogroups in which every strain has at least 1 gene copy.** Support values show the proportion of gene trees that are congruent with the focal node and blue boxes indicate resolved monophyletic species clades identified by multilocus species delimitation using tr2 (see Methods). Shaded boxes refer to the *F. fujikuroi* (yellow) and *F. oxysporum* (orange) species complexes. Scale bar indicates 0.02 substitutions per site. The *F. xylarioides* strain types arabica, robusta, and coffea are labelled. *Fx*, *F. xylarioides*; *Fo*, *F. oxysporum*. Full strain details in S1 and S2 Tables. The data underlying this figure can be found in https://doi.org/10.5281/zenodo.13836286.
(TIFF)

**S3 Fig. Whole-genome similarity between *Fusarium oxysporum* and *Fusarium fujikuroi* species complexes.** Cells indicate with colour the nucleotide similarity across all predicted coding sequences between the strains on the x and y axes. Shaded boxes refer to: the *Fusarium oxysporum* species complex, orange; and the *Fusarium fujikuroi* species complex, yellow, of which *Fusarium xylarioides* is a member. Arabica, robusta, coffea1, and coffea2 genomes all belong to *Fusarium xylarioides*. Excluding *F. xylarioides*, one genome is shown for each species/*formae speciales* with strain details in S2 Table. Fo, *F. oxysporum*. The data underlying this figure can be found in https://doi.org/10.5281/zenodo.13836286.
(TIFF)

**S4 Fig. Gene sharing across the *Fusarium xylarioides* host-specific populations.** Orthogroups shared between *F. xylarioides* arabica, robusta, and the 2 coffea clusters. Drawn 16,006 (excluding 10,567 that were absent from *F. xylarioides*.
(TIFF)

**S5 Fig.** Whole-genome alignments (A) *Fusarium verticillioides* chromosomal assembly mapped against the *Fusarium xylarioides* arabica563 contig assembly and (B) *Fusarium xylarioides* robusta254 mapped against the *Fusarium xylarioides* arabica563 contig assembly. (A) Each bar represents one of the 16 arabica563 contigs >100 kb in length that are also present in *Fusarium verticillioides*, with contig size and position on the x axis. Arabica563 contig numbers are annotated in-line with the bars. Shaded regions indicate the presence of syntenic *Fusarium verticillioides* chromosomes, the colour of which identifies the specific mapped chromosome. Unshaded regions indicate those found only in arabica563, and arabica563 contig 16 is absent from *Fusarium verticillioides*. *Fusarium verticillioides* chromosomes CM010967.1–CM010977.1 correspond to 1–11 in the genome accession GCA_000149555.1. (B) Each bar represents one of the 16 *Fusarium xylarioides* arabica563 contigs that are also present in *Fusarium xylarioides* robusta254, with contig size and position on the x axis. Shaded regions indicate the presence of syntenic *Fusarium xylarioides* robusta254 contigs, the colour of which identifies the specific mapped contig from *Fusarium xylarioides* robusta254.

Unshaded regions indicate contigs found only in *Fusarium xylarioides* arabica563 that are absent in *Fusarium xylarioides* robusta254. Telomeric repeats are annotated as blue diamonds on their corresponding scaffolds.
(TIFF)

**S6 Fig. PHYML phylogenetic trees for HTRs 3, 4, 5, and 6.** All trees are drawn with 100 bootstraps, branch labels reflect bootstrap support values. Tip labels correspond to NCBI GenBank accession numbers or *F. xylarioides* chromosmes and are shaded by their *F. xylarioides* population or species complex. Scale bar shows genetic distance. A HTR 3 on chromosome 8 B HTR 4 on chromosome 11 C HTR 5 on supercontig 2 D HTR 6 in robusta, coffea1 and coffea2 genomes. FFC, *F. fujikuroi* complex; FOSC, *F. oxysporum* species complex. The data underlying this figure can be found in https://doi.org/10.5281/zenodo.13836286.
(TIFF)

**S7 Fig. Different parts of *Fusarium oxysporum* f. sp. *lycopersici* supercontig 51 is shared with *Fusarium xylarioides* populations.** A region of supercontig 51 in *Fusarium oxysporum* f. sp. *lycopersici* chromosome 14 (NC 030999.1, middle bar) is shared which matches to HTR 2 on arabica563 chromosome 5 (bottom plot, grey rectangle indicates the region which is shared). The shared genes in this 4-kb region are all secreted in xylem (*SIX*) effector genes, which are shaded according to the legend and labelled. A different part of supercontig 51 is shared with *F. xylarioides* robusta genomes and matches to a scaffold which is not in the arabica563 reference genome. Repeats of interest are shaded according to the legend and unclassified repeats which occur once are labelled "rnd." The locations for six12 and six7 are from [25]. Figure drawn to scale. (B) Heatmap showing presence of orthologous groups found in the region found in *F. xylarioides* robusta from (A), compared with *Fusarium* phylogeny. Present genes are shaded in dark and all share >96% identity. The phylogenetic species tree was created following S3 Fig. Full strain details in Tables S2 and 1. The data underlying this figure can be found in https://doi.org/10.5281/zenodo.13836286.
(TIFF)

**S8 Fig. Putative horizontal transfer 1 (HTR 1) between *Fusarium oxysporum* and *Fusarium xylarioides*.** (A) HTR 1 on chromosome 1 in *Fusarium xylarioides* arabica563 (544 kb—739 kb, bottom plot) matches 2 regions in *Fusarium oxysporum* f. sp. *lycopersici* chromosome 3 (NC 030988, 4.7 Mb—5.6 Mb, top plot) in 2 pieces, large (200 kb) and small (20 kb). Highly similar genes shared by the 2 species are shaded in purple in top and bottom plots, a subset are labelled by *Fusarium oxysporum* gene locus tag, and non-matching genes found in the shared region are shaded grey. Genes flanking the shared region in *Fusarium oxysporum* are shaded yellow, and those which are found in *Fusarium xylarioides* (BLAST length >80%) have a lower sequence identity and are absent from contig 13. (B) A DendroBLAST rooted gene tree for the arabica563 gene H9Q71 0002169 (OG0015707) orthologous group, found in the shared region. (C) DendroBLAST gene tree for the arabica563 gene H9Q71 0003005 (OG0017996) orthologous group, found in the shared region. (D) The genes shared between the 2 species in the matching region have a higher BLAST sequence identity (mean 94%, dotted line) than those found in the flanking regions either side (mean 88%, dotted line). Figure drawn to scale. The data underlying this figure can be found in https://doi.org/10.5281/zenodo.13836286.
(TIFF)

**S9 Fig. Differential expression of genes and putative effector genes in *Fusarium xylarioides*.** Genes up-regulated in the in planta samples are shaded orange and those down-regulated are shaded blue. Putative effectors identified in [28] are labelled according to their class: yellow, pre-described fungal plant pathogens; purple, small and cysteine-rich proteins; blue,

carbohydrate-active enzymes (S6 and S7 Tables). The data underlying this figure can be found in https://doi.org/10.5281/zenodo.13836286.
(TIFF)

**S10 Fig. Whole-genome view of the loci of transposable elements in the *F. xylarioides* arabica563 reference genome.** Transposable elements shown are those which were newly identified on the mobile and pathogenic chromosome of *F. oxysporum* f. sp. *lycopersici* by [25], as well as the unclassified TE "rnd-6 family-1942." Chromosomes are labelled with *F. xylarioides* contig numbers (see S12 Table).
(TIFF)

**S11 Fig. Miniature *impala* (*mimp*) family 1 sequence genealogy.** All *mimp* family 1 sequences from Fig 6 were aligned using MAFFT in Geneious v9.1.8. Using BLASTn, the consensus sequence was used to find copies in all other genomes used in this study (S2 and in the nr database. Any species with a hit >80% length and identity were selected (additional *F. xylarioides* and *F. oxysporum* hits were ignored), all sequences were re-aligned and a maximum likelihood tree with 1,000 bootstrap replicates constructed with IQ-TREE2. Bootstrap branch support is shown for all branches >0.7 (calculated with 1,000 bootstraps). Each point corresponds to a *mimp* sequence and the scalebar represents 0.1 substitutions per site. Tree tip labels are shaded by phylogenetic group: arabica, robusta, coffea1, and coffea2 *F. xylarioides* populations; FFC, *F. fujikuroi* species complex (all FFC tips are *F. proliferatum*; FOSC, *F. oxysporum* species complex. *F. oxysporum* strains are those annotated in this Fig 6). The data underlying this figure can be found in https://doi.org/10.5281/zenodo.13836286.
(TIFF)

**S1 Table. Bio-geographic details for strains sequenced in this study.**
(PDF)

**S2 Table. Published genomes analysed in this study.** The columns describe: species, short species with strain number, accession number, whether the genome was used in OrthoFinder, whether the genome was used in S3B Fig for whole-genome similarity (Nsimscan).
(PDF)

**S3 Table. Population genetics statistics related to genetic differentiation ($d_{xy}$) in four *Fusarium xylarioides* populations.** Measured in 100 kb windows.
(PDF)

**S4 Table. Nineteen up-regulated genes in both Fusarium xylarioides arabica strains are absent from the Fusarium fujikuroi complex (FFC) and differentially present across Fusarium xylarioides, the Fusarium oxysporum species complex (FOSC), the Fusarium graminearum species complex (FGC), and the Fusarium solani species complex (FSC).** Species presence was determined using genealogies from OrthoFinder (see Methods) and BLAST similarity. A caret indicates a genealogy present in Fusarium udum, the other vascular wilt in the Fusarium fujikuroi species complex. Partial shading indicates partial presence across the population. An asterisk represents genealogies which match Fusarium oxysporum effector proteins.
(PDF)

**S5 Table. Predicted effectors in arabica563 genes by EffectorP [34].** Only those which were up-regulated in both *Fusarium xylarioides* arabica563 and arabica908 are shown. The numbers in brackets describe the certainty. The orthogroup is shown for those predicted effectors up-regulated in both arabica563 and arabica908, and those which are annotated as carbohydrate-active enzymes are described. H9Q71 0010006 encodes an Ecp2 effector protein and H9Q71

0007533 contains a LysM domain.
(PDF)

**S6 Table. Expression data for all putative effectors from [28] across the *Fusarium xylarioides* arabica563 samples.** Each gene is described as up-regulated in planta ("coffee.up"), in axenic ("culture.up"), or not differentially expressed ("ns").
(PDF)

**S7 Table. Expression data for all putative effectors from [28] across the *Fusarium xylarioides* arabica908 samples.** Each gene is described as up-regulated in planta ("coffee.up"), in axenic ("culture.up"), or not differentially expressed ("ns").
(PDF)

**S8 Table. The number of genes expressed for each carbohydrate-active enzyme subfamily in *Fusarium xylarioides* arabica563.** The in planta up-regulated genes are shaded, where the warmest colours represent the highest up-regulated gene count.
(PDF)

**S9 Table. The number of genes expressed for each carbohydrate-active enzyme subfamily in *Fusarium xylarioides* arabica908.** The in planta up-regulated genes are shaded, where the warmest colours represent the highest up-regulated gene count.
(PDF)

**S10 Table. Gene count and description for significantly enriched InterPro terms in the *Coffea arabica* in planta samples.**
(PDF)

**S11 Table. Transposable element density in horizontally transferred regions (HTR), *Starships* and genome-wide in the *Fusarium xylarioides* arabica563 reference.** Low complexity and simple repeats were excluded from the transposable element (TE) count. "TE of interest" refers to those transposable elements identified on the *F. oxysporum* mobile pathogenic chromsome in [25] and the unclassified TE "rnd-6 family-1942." sc2, supercontig 2.
(PDF)

**S12 Table. The contigs in the *Fusarium xylarioides* arabica563 reference genome which correspond to each chromosome in the *F. verticillioides* assembly [21].** The second column represents the contigs as named in the public assembly under accession PRJNA1043203. In the final column, each contig has been renamed to first correspond to its core *Fusarium* chromo-some, and then its order in the whole genome visualisation (Fig 1A).
(PDF)

**S13 Table. Comparison of divergence patterns for HTR regions to those of background genome regions.** Mean bedtools coverage in 10 kb regions relative to the whole genome. $I\_Fx$ = mean similarity with other *F. xylarioides* populations across 10 kb windows within HTR. $I\_Fo$ = mean similarity with closest *F. oxysporum* match across 10 kb windows within HTR. $I\_Diff$ is $I\_Fx - I\_Fo$, a metric of how similar is *F. xylarioides* versus *F. oxysporum*: positive value = more similar to *F. xylarioides*; negative = more similar to *F. oxysporum*. $p$ = proportion of values of $I\_Diff$ from all contiguous regions of genome the same size as HTR that are $\leq$ the observed value of $I\_Diff$ for the HTR.
(PDF)

**S14 Table. Comparison of divergence patterns for HTR regions to those of background genome regions.**

Phylogenetic divergence relative to single copy ortholog gene trees. *D_Fx* = mean pairwise divergence among *F. xylarioides* genomes for given HTR (substitutions. per site). *D_Fo* = mean pairwise divergence among *F. oxysporum* genomes for a given HTR. *D_FxFo* = minimum pairwise divergence between *F. xylarioides* HTR and closest *F. oxysporum* match. p = proportion of values of *D_FxFo* across single ortholog gene trees ≤ the observed value of *D_FxFo* for the HTR. Mean *D_FxFo* across single ortholog gene trees = 0.067, 95% range 0.024–0.153 substitutions per site.
(PDF)

**S1 File. Genomic loci for the 6 putative horizontally transferred regions and the *F. xylarioides* strain in which they were identified.** This file can be opened in a text editor program.
(BED)

**S2 File. Nucleotide sequences for the 6 putative horizontally transferred regions from the *F. xylarioides* strain in which they were identified.** This file can be opened in a text editor program.
(FA)

**S3 File. Predicted gene annotations and functions for *Starship* cargo and those annotations which flank the *Starship*.**
(XLSX)

**S4 File. Sequence read archive (SRA) accessions and data counts for filtered data.**
(XLSX)

## Acknowledgments

We thank W. Gordon, J. Meyer, M. Rutherford, T. Kibani, G. Armstrong, D. Dring, R. Hillocks, E. Khonga, and R. Mehrotra, as well as other unnamed collectors who originally isolated the fungal pathogens and deposited them in the CABI-IMI collection, and 4 reviewers. With thanks to the Imperial College Stevenson Fund for sponsoring L. D. Peck's research placement in Génétique et Écologie Évolutives at Université Paris Saclay, and to "i Focus & Write." With special thanks to A. Snirc for sequencing genomes, J. Vernadet for help with variant-calling, A. van Westerhoven for providing the *F. oxysporum* f. sp. *cubense* reference genome, E. Gluck Thaler and A. Vogan for *Star*-fishing, and S. Gurr and M. Fisher for valuable comments.

## Author Contributions

**Conceptualization:** Lily D. Peck, Matthew J. Ryan, Timothy G. Barraclough.

**Data curation:** Lily D. Peck.

**Formal analysis:** Lily D. Peck, Theo Llewellyn, Bastien Bennetot, Samuel O'Donnell, Reuben W. Nowell, Ricardo C. Rodríguez de la Vega, Jeanne Ropars, Timothy G. Barraclough.

**Funding acquisition:** Lily D. Peck, Matthew J. Ryan, Tatiana Giraud, Timothy G. Barraclough.

**Investigation:** Lily D. Peck.

**Methodology:** Lily D. Peck.

**Resources:** Matthew J. Ryan.

**Supervision:** Matthew J. Ryan, Julie Flood, Tatiana Giraud, Pietro D. Spanu, Timothy G. Barraclough.

**Validation:** Lily D. Peck.

**Visualization:** Lily D. Peck.

**Writing – original draft:** Lily D. Peck, Timothy G. Barraclough.

**Writing – review & editing:** Lily D. Peck, Theo Llewellyn, Reuben W. Nowell, Matthew J. Ryan, Julie Flood, Ricardo C. Rodríguez de la Vega, Jeanne Ropars, Tatiana Giraud, Pietro D. Spanu, Timothy G. Barraclough.

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
