## [Editor Report · Decision Letter 0]

19 Dec 2023

Dear Dr Peck, 

Thank you for submitting your manuscript entitled "Horizontal transfers between fungal Fusarium species contributed to successive outbreaks of coffee wilt disease" for consideration as a Research Article by PLOS Biology.

Your manuscript has now been evaluated by the PLOS Biology editorial staff as well as by an academic editor with relevant expertise and I am writing to let you know that we would like to send your submission out for external peer review.

Once your full submission is complete, your paper will undergo a series of checks in preparation for peer review. After your manuscript has passed the checks it will be sent out for review. To provide the metadata for your submission, please Login to Editorial Manager (https://www.editorialmanager.com/pbiology) within two working days, i.e. by Dec 21 2023 11:59PM.

Kind regards,

Melissa

Melissa Vazquez Hernandez, Ph.D.

Associate Editor

PLOS Biology

---

## [Decision Letter · Decision Letter 1]

5 Feb 2024

Dear Dr Peck,

Thank you for your patience while your manuscript "Horizontal transfers between fungal Fusarium species contributed to successive outbreaks of coffee wilt disease" was peer-reviewed at PLOS Biology. I apologized for the long waiting time. Your manuscript has been evaluated by the PLOS Biology editors, an Academic Editor with relevant expertise, and by several independent reviewers.

As you will see in the reviewer reports, while there is recognition of the potential significance of your findings, a significant number of crucial concerns have been raised. Following discussions with the Academic Editor, it is evident that substantial revisions would be necessary to meet the criteria for publication in PLOS Biology. Given our and the reviewers' interest in your study, we are open to considering a comprehensive revision; however, this would require thorough addressing of all reviewers' comments. A successful revision must address issues such as strengthening the HTR hypothesis, correcting the inappropriate use of FST, incorporating available genomic data, including bootstrap support in the phylogenetic analysis, providing support for MIMPS, and enhancing statistical support. Additionally, the revision should clarify the lineage sorting hypothesis and thoroughly discuss it.

IMPORTANT: Please note that the Academic Editor has kindly provided some additional guidance as to what needs to be addressed before we can consider the study further (see the foot of this email). All other concerns raised by the reviewers should also be addressed.

Given the extent of revision that would be needed, we cannot make a decision about publication until we have seen the revised manuscript and your response to the reviewers' comments. Your revised manuscript would need to be seen by the reviewers again, but please note that we would not engage them unless their main concerns have been addressed.

We appreciate that these requests represent a great deal of extra work, and we are willing to relax our standard revision time to allow you 6 months to revise your study. Please email us (plosbiology@plos.org) if you have any questions or concerns, or envision needing a (short) extension.

**IMPORTANT - SUBMITTING YOUR REVISION**

*Resubmission Checklist*

*Published Peer Review*

*PLOS Data Policy*

*Blot and Gel Data Policy*

Sincerely,

Melissa

Melissa Vazquez Hernandez, Ph.D.

Associate Editor

PLOS Biology

COMMENTS FROM THE ACADEMIC EDITOR:

The critical issues that need addressing, in my opinion, are:

The available genome datasets that weren't used, especially the more comprehensive chromosome-level assemblies and the species that were excluded.

The problem of lineage sorting. It's unacceptable for this to be mentioned only once and in an ambiguous manner (page 7), leaving room for this alternative hypothesis without further detail. This needs to be directly tackled (as even reviewer 2 suggested) and discussed more thoroughly in the paper. Personally, I'm skeptical of horizontal gene transfer (HGT/HTR) claims when the boundaries of the transferred segment are undefined.

Reviewer 1 expressed several important criticisms, notably regarding the inappropriate use of FST.

Reviewer 2 criticized the bootstrap support for the phylogenetic trees and the classification into four clades, which is a fundamental issue. Reviewer 3 also highlighted the inadequate bootstrap support, which is really concerning as this is so fundamental to these analyses.

REVIEWER'S COMMENTS:

— — — — — 

Reviewer #1: 

In this manuscript the authors investigate an hypothesis that was proposed previously that there are some regions of the genomes of F. xylarioides were potentially as the result of horizontal gene transfer (HTR) from another Fusarium species, F. oxysporum. To do this additional genomes have been sequenced, both F. xylarioides and F. oxysporum. In addition the authors interrogated whether genes in the putative HTR regions were highly expressed during infection of plants by two of the isolates studied.

Overall I found that the manuscript seems to consist of two stories, one is about the differences and similarities between F. xylarioides isolates, which is in itself interesting and useful with regards to managing these pathogens. The second is the hypothesis that some of genome segments in F. xylarioides seem to have originated from another source and share significant similarity with equivalent sequences in F. oxysporum

The authors make a convincing argument with regards to HTR but I miss a number of aspects. The first is that there is a long read genome for F. xylarioides and while they suggest that a full chromosome assembly was not possible I think that it would be useful to perhaps assign some of the contigs to chromosomes. Most species in the Fusarium fujikuroi species complex have 12 "core" chromosomes and then some have accessory chromosomes. Given that there are chromosome assemblies of some related species this assignation should be possible?

Different strain types featured different putative effector genes, whereby some effectors showed close matches with F. oxysporum, but were absent from closer relatives of F. xylarioides. But the authors do not seem to have considered many other species that are in the Fusarium fujikuroi species complex - and there are many more genomes available for comparison than those that are used in this study. This would be an easy comparison to make and potentially strengthens the argument for HTR.

It is also not clear why the authors chose to sequence the genomes of F. oxysporum isolates in this study (I understand the rationale in terms of which forma specialis were chosen) given that there are many genomes already available including (with one exception) the forma specialis which were chosen for this study. These other genomes could also be considered in this analysis for the presence of the HTR regions - if only to show that these are not isolate specific in F. oxysporum?

An aspect of this study that is discussed is that there are in fact potentially more than one species within the F. xylarioides complex. I think that the authors show good evidence that there are in fact at least two species and to consider these just different populations is confusing and not strictly accurate. I think that this aspect does make understanding this study more complicated. The fact that the authors "rename" the isolates arabica, robusta and coffea actually speaks volumes to the fact that these are probably three different species. 

I do not agree that one can reasonably do an FST calculation on such a small number of isolates.

In supplementary figure 1 the asterix is not explained?

The names of the genomes and isolates is very complicated and I found myself getting very confused. For example in figures 3 and 6 the isolatename I039 appears but is not in any of the tables, I had to go back to the previous publication to understand this and then noted that this isolate name is in fact in the supplementary section named as I-394. Also there is the F. verticilliodes isolate that is alternatively 14953 and also 953. The naming's of the isolates is different in supplementary figure 1 and 2 (for some reason in the supplementary table 2 F. xylarioides 379925 and L0394 are written in italics) I suggest that a single table providing all the names of all the isolates for which genomes have been used in the study is necessary. In addition the same name needs to be used for a particular isolate throughout the entire manuscript (the supplementary table 2 goes some way to providing this information but then does not include the isolates sequenced in this study). This way when one compares figures it is possible to easily understand what is going on.

One of the most perplexing aspects of this manuscript is that in some cases analyses are done using all the genomes and in others the authors have focused only on the genomes that were sequenced in this study. This is rather unbalanced and does not seem to be needed?

The last sentence of the abstract states "Our results support the hypothesis that horizontal gene transfers contributed to the repeated emergence of this fungal disease." This sentence is problematic because I think it refers to F. xylarioides but the last species referred to in the previous sentence is in fact F. oxysporum. Another aspect is that while fungi can be causal agents of a plant disease, one cannot strictly refer to them as causing a fungal disease. Surely a fungal disease is a disease of a fungus?

— — — — — 

Reviewer #2: This tour de force by Peck et al seeks to elucidate how horizontal transfers between Fusarium species has contributed to the outbreaks of devastating coffee wilt disease. I must commend the authors on such a thorough study, and I must say I enjoy reading it. Therefore, the comments I have seek to improve the overall study and tie up a couple of loose ends.

My main comments on this is that the foundational analyses and findings need a little more support. The follow on findings will then be bolstered by this. Primarily, the authors state in the Results that 'there are four well resolved groups', yet my issue with this is that there isn't basic bootstrap support on the phylogenetic analysis to support this. Then, when talking about coffea strains falling into two distinct clades, again, bootstrap support would help here. For these two clades it would be worth noting what makes them distinct, i.e. are there differences in gene content (this may have been done and I didn't see it, in which case, make it a bit clearer). 

Perhaps my biggest comments are:

- does temporal analysis using BEAST/BEAST2 confirm your findings? This may be beyond the scope of this study, but would be nice.

- I think there needs to be a bit more consideration of recombination and/or gene flow, which would be resolved by including an ADMIXTURE and/or fineSTRUCTURE analysis. Indeed, the latter would help resolve ancestral donation and inferring the 'donor strains'.

Minor comments:

Data availability: for me, I get the notification that there's been no data deposited or its awaiting submission/validation.

Introduction: a couple of sentences describing the yield losses (amounts/percentage) attributed to fungi would help non-Fusarium/fungal specialist readers. Also, perhaps expanding the recent disease severity in Ethiopia. Is this due to climate change? Also in the introduction, it should be noted that Starships haven't been proven to be involved in HGT (i.e. no one has 'seen' them move) - see Bucknell et al 2023 Mol Microbiology. At the moment, it's all sequence evidence, which is fine, but we are not completely 100% sure of how they are moving.

Methods: 'other Fusarium genomes' - more details needed in the text on the sequence read archive or refer to Table 1. Is the TE/repeat annotation library available for community use? Importantly, how were orthologous gene group trees rooted (needs mention in Methods)? How does your Starship identification align with the 'official' starfish method for discovering these?

Figures: a description of what the scale is describing is needed in the legend for all phylogenies. 

I couldn't find Table S1!

— — — — — 

Reviewer #3: The manuscript by Peck et al. analyzes the outbreaks caused by the coffee wilt disease agent Fusarium xylarioides. The authors take advantage of historical samples for a total of 13 strains to investigate the genetic structure and host associations. The authors indicate that virulence was acquired at least in part through horizontal transfer of effector genes. Roles in virulence were supported by transcriptomic analyses during infection.

The manuscript reports additional genomes for Fx over existing historical genome datasets and provides new dimensions of data analyses. However, much of the major claims of horizontal gene transfer and virulence components are not supported by robust analyses allowing for hypothesis testing. Hence, most of the manuscript remains largely descriptive and suggestive. The main claim stated in the title and elsewhere that horizontal transfer contributed to outbreaks, even successive outbreaks, lacks support on multiple levels.

Evidence for HTRs: the authors describe "putative HTRs" by identifying similar genomic regions between F. oxysporum and F. xylarioides. I am struggling to understand how the evidence for this was evaluated. Neither the main results nor Figure 4 provide direct support. Trees should include bootstrap values. 

As Fo and Fx are closely related, how was the alternative explanation of incomplete lineage sorting evaluated? 

The results should also more clearly state how cut-offs for similarity were defined using genome-wide average divergence, i.e. how much more similar are putative HTRs compared to the genome-wide average divergence? 

Was similarity restricted to coding sequences or did similarity expand across the entire HTR including non-coding sequences? The issue of similarity analyses extends also to the Six gene analyses where it remains unclear whether the reported similarities are true outliers compared to expectations. Six genes are expected to diverge more rapidly. What divergence or similarity is hence considered "unusual", falling outside of the expectation for vertical inheritance? Also for HTR 4, the authors report 88% flanking sequence identity, 90% genome-wide identity and >94% defined as "high" identity. Are these true outliers / what percentiles? 

Do Starship regions suspected to be HTR share higher similarity than other regions? Can you provide more insights into the "overlap" of a Starship and HTR3? What is the suspected mechanism? That the Starship was the mechanism of HT?

Overall, without proper statistical support, it is difficult to discussing evidence for HTR in this system.

Upregulation of CAZymes and predicted effectors: This section suggests that there is an overlap between gene upregulation and candidate genes involved in virulence. I lack though a clear conclusion from the section. Was the aim to show that effector candidates tend to be upregulated in planta? Were effector candidates more likely to be upregulated? Furthermore, how is the statement "It appears that a subset of CAZyme families have been acquired..." related to the HTR section above? What HTR region is this? How strong is the evidence for HT there? Is the pattern of upregulation conserved among the homologous segments in the putative donor and recipient strain genomes?

Analyses of miniature impala elements and HTR: Here I also lack a clear statistical support whether e.g. mimps are more likely to be putatively horizontally transferred. The authors simply refer to the fact that some mimps were found in HTRs. 

Is "shared" TEs between Fx and Fo to be understood as being horizontally transferred or shared by recent, common descent? What statistical analyses support either of these interpretations? 

Additional comments:

- Figure 3B: The authors attribute the small amount of reticulation in the network to gene flow. However, this could be explained by a lack of resolution of the network and/or historic recombination. Later this possibility is mentioned Clarifying the reticulation pattern would be helpful.

- FST analyses among genetic groups seem inappropriate as it remains unclear whether each group indeed consists of a population of interbreeding individuals (a requirement for calculating FST). A group size of ~2-4 haploid individuals is also inappropriate to estimate allele frequencies (necessary for FST calculations).

- Discussing the existence of "populations" with such small group sizes of genetic clusters seems tenuous. Without further sampling efforts, terms like "lineage" or "genetic cluster" seem more appropriate.

- The Circos plot identifies identities spanning 0-1. What is the interpretation of similarities of e.g. 0.2? Isn't 20% similarity at the level of random expectation for sequence similarity? Similarities would typically only be reported for syntenic or aligned regions. So 70-80% might be the lower limit for this. Clarifying this in the legend would be helpful.

— — — — — 

Reviewer #4: This manuscript "Horizontal transfers between fungal Fusarium species contributed to successive outbreaks of coffee wilt disease" investigate the coffee wilt disease caused by the fungal pathogen Fusarium xylarioides focusing on the sub-Saharan Africa area. Authors sequenced and compared 13 genomes, including 11 Fx, F6 o and Fs, generated one Fx genome using long-read to be used as the Fx reference genome, and conducted comparative genomics and transcriptomics analysis. This is an important research topic. 

Authors have done tremendous work, trying to answer at least 3 rather complex biological questions: 1) populations structure of Fx isolates 2) HGT, and 3) mechanisms of HGT. Unfortunately, I am not able to derive the same conclusions authors presented here based on the data presented. I believe more careful analysis that testing specific hypothesis proposed in the manuscript is necessary. 

Specific comments. 

First, authors concluded that F. xylarioides comprises at least four distinct lineages: one host-specific to Coffea arabica, one to C. canephora var. robusta, and two historic lineages isolated

from various Coffea species. The data presented in the manuscript doesn't support this. 

* Figure 2a supports the monophyletic nature of the Fx isolates. 2b shows no differences among F. x genomes. Current phylogenetic analyses, including Figure 3 didn't provide strong evidence supporting the distinct affiliation with different coffee hosts. 

* In addition, Figure 2 should use the single-copy genes instead of genes have members in all genome. Bootstrap values are needed on the phylogeny. 

* Also a detailed table capture all Fx strains included in the analysis, not only sequenced strains, may help to see the relationships among these strains from both geological and historical perspectives. 

Second: authors concluded that host-specificity appears to be acquired through horizontal transfer of effector genes from members of the F. oxysporum species complex by comparing a F. oxysporum tomato pathogen to the Fx coffee pathogen and identified regions on contigs 3, 6, 10, 11 and 13. 

* Why not the F. oxysporum coffee pathogen? 

* As presented in Figure 1 and stated in the manuscript, "HTRs were differentially present across the F. xylarioides populations". If these regions are important for Fx pathogenicity to coffee, why they are so different among different Fx isolates? 

* Do these present/absent polymorphisms have any relationship to the Fx populations structure?

Third, authors presented the affiliation of Mimps between F. oxysporum genome and Fx. And concluded that facilitate HGT. Turning affiliation into facilitation is too big a leap. 

Overall, authors added unsupported speculation in the discussion section. For instance, authors claim that "the F. xylarioides formae speciales proposed nearly two decades ago can now be recognised as genetically differentiated populations with different host-specificity". In the result section, authors stated that there are 4 distinct populations. Then

which strains will be classified as F. xylarioides f. sp. abyssiniae and F. xylarioides f. sp. canephorae, respectively?

---

## [Decision Letter · Decision Letter 2]

11 Sep 2024

Dear Dr Peck,

Thank you for your patience while we considered your revised manuscript "Horizontal transfers between fungal Fusarium species contributed to successive outbreaks of coffee wilt disease" for publication as a Research Article at PLOS Biology. This revised version of your manuscript has been evaluated by the PLOS Biology editors, the Academic Editor and the original reviewers, being Brenda Wingfield reviewer #1.

Based on the reviews and on our Academic Editor's assessment of your revision, we are likely to accept this manuscript for publication, provided you satisfactorily address the remaining points raised by the reviewers, however it is not necessary to perform the additional analyses suggested by Reviewer #3. Please also make sure to address the following data and other policy-related requests.

a) We would strongly recommend you to publish the reports. This would be an option once the study is accepted.

b) We do not have a word limit and we believe having all material and methods in the main text allows them to be more accessible for readers. Please place the Supplementary Material and Methods in the main text.

Please supply the numerical values either in the a supplementary file or as a permanent DOI’d deposition for the following figures:

Figure 3BC, 5, 6B, S3, S6ABC, S7B, S8D, S9

d) Please cite the location of the data clearly in all relevant main and supplementary Figure legends, e.g. “The data underlying this Figure can be found in S1 Data” or “The data underlying this Figure can be found in https://doi.org/10.5281/zenodo.XXXXX”

e) Please provide the tree files for Figures Fig 1C, 2, 3BC, 6A, 7B, S1, S2, S6ABCD, S7B, S8BC, S11

f) Please ensure that your Data Statement in the submission system accurately describes where your data can be found and is in final format, as it will be published as written there.

g) Per journal policy, if you have generated any custom code during the course of this investigation, please make it available without restrictions upon publication. Please ensure that the code is sufficiently well documented and reusable, and that your Data Statement in the Editorial Manager submission system accurately describes where your code can be found.

We expect to receive your revised manuscript within two weeks. 

*Published Peer Review History*

*Press*

Sincerely,

Melissa

Melissa Vazquez Hernandez, Ph.D.

Associate Editor

PLOS Biology

REVIEWERS' COMMENTS:

Reviewer #1: 

please correct in the supplementary table 2 - F. xylarioides 379925 and L0394 are written in italics

Reviewer #2: 

I have no comments on the revision - all my comments were addressed in the previous revision round. I recommend accept

Reviewer #3: 

The authors have conducted extensive new analyses and re-analyses in an attempt to support many of the claims made in the previous version. 

Notable improvements include analyzing a more complete set of the already existing genomes and evaluating alternative hypotheses to the claim of horizontal gene transfer. 

The new version now also lacks several analyses that were not replaced with more appropriate investigations (e.g. population differentiation analyses, etc.).

The evidence for horizontal gene transfer is now clearly and more transparently laid out. The authors primarily can show that the regions labeled as HT are outliers in the genome for their phylogenetic signal (or similarity patterns), which is consistent with HT. Rare incomplete sorting would also be an outlier obviously. I take the finding of a Starship in one of the HT regions as an encouraging sign though. This would indeed provide a mechanism for how DNA of common descent can appear in distinct lineages outside of vertical transmission.

Finally, I remain confused about what "identities" spanning from 0-1 represent. The tool that was used analyzes mapped read coverage but not what is typically called "sequence identity" or homology. If sequence coverage is shown by the 0-1 range, then the term "identity" is wrong. This would simply reflect that there were alignable reads without any statement about how perfectly this read aligned (i.e- how identical the two genomes are at this locus).

---

## [Editor Report · Decision Letter 3]

30 Sep 2024

Dear Dr Peck,

Thank you for the submission of your revised Research Article "Horizontal transfers between fungal Fusarium species contributed to successive outbreaks of coffee wilt disease" for publication in PLOS Biology. On behalf of my colleagues and the Academic Editor, Sophien Kamoun, I am pleased to say that we can in principle accept your manuscript for publication, provided you address any remaining formatting and reporting issues. These will be detailed in an email you should receive within 2-3 business days from our colleagues in the journal operations team; no action is required from you until then. Please note that we will not be able to formally accept your manuscript and schedule it for publication until you have completed any requested changes.

PRESS

Sincerely, 

Melissa

Melissa Vazquez Hernandez, Ph.D., Ph.D.

Associate Editor

PLOS Biology
